# The DEAD-box RNA helicase CshA is required for fatty acid homeostasis in *Staphylococcus aureus*

**Vanessa Khemici, Julien Prados, Bianca Petrignani[¤a], Benjamin Di Nolfi, Elodie Bergé, Caroline Manzano, Caroline Giraud[¤b], Patrick Linder***

Department of Microbiology and Molecular Medicine, Faculty of Medicine, University of Geneva, Geneva, Switzerland

¤a Current address: Global Health Institute, School of Life Sciences, Ecole Polytechnique Federale de Lausanne, Lausanne, Switzerland.
¤b Current address: Normandie Univ, UNICAEN, Caen, France
* patrick.linder@unige.ch

**Data Availability Statement:** The data are available at: https://www.ncbi.nlm.nih.gov/geo/query/acc.cgi?acc=GSE133013.

## Abstract

*Staphylococcus aureus* is an opportunistic pathogen that can grow in a wide array of conditions: on abiotic surfaces, on the skin, in the nose, in planktonic or biofilm forms and can cause many type of infections. Consequently, *S. aureus* must be able to adapt rapidly to these changing growth conditions, an ability largely driven at the posttranscriptional level. RNA helicases of the DEAD-box family play an important part in this process. In particular, CshA, which is part of the degradosome, is required for the rapid turnover of certain mRNAs and its deletion results in cold-sensitivity. To understand the molecular basis of this phenotype, we conducted a large genetic screen isolating 82 independent suppressors of cold growth. Full genome sequencing revealed the fatty acid synthesis pathway affected in many suppressor strains. Consistent with that result, sublethal doses of triclosan, a FASII inhibitor, can partially restore growth of a *cshA* mutant in the cold. Overexpression of the genes involved in branched-chain fatty acid synthesis was also able to suppress the cold-sensitivity. Using gas chromatography analysis of fatty acids, we observed an imbalance of straight and branched-chain fatty acids in the *cshA* mutant, compared to the wild-type. This imbalance is compensated in the suppressor strains. Thus, we reveal for the first time that the cold sensitive growth phenotype of a DEAD-box mutant can be explained, at least partially, by an improper membrane composition. The defect correlates with an accumulation of the pyruvate dehydrogenase complex mRNA, which is inefficiently degraded in absence of CshA. We propose that the resulting accumulation of acetyl-CoA fuels straight-chained fatty acid production at the expense of the branched ones. Strikingly, addition of acetate into the medium mimics the *cshA* deletion phenotype, resulting in cold sensitivity suppressed by the mutations found in our genetic screen or by sublethal doses of triclosan.

**Funding:** This work was supported by the University of Geneva (http://www.unige.ch) and the Swiss National Science Foundation to PL (grants 149228, 170207, 188736; http://www.snf.ch) to PL. The funders had no role in study design, data collection and analysis, decision to publish, or preparation of the manuscript.

**Competing interests:** The authors have declared that no competing interests exist.

## Author summary

DEAD-box RNA helicases are highly conserved proteins found in all domains of life. By acting on RNA secondary structures they determine the fate of RNA from transcription to degradation. Bacterial DEAD-box RNA helicases are not essential under laboratory conditions but are required for fitness and under stress conditions. Whereas many DEAD-box protein mutants display a cold sensitive phenotype, the underlying mechanisms have been studied only in few cases and found to be associated with ribosome biogenesis. We aimed here to elucidate the cold sensitivity of a *cshA* mutant in the Gram-positive opportunist pathogen *Staphylococcus aureus*. Our study revealed for the first time that part of the cold sensitivity is related to the inability of the bacterium to adapt the cytoplasmic membrane to lower temperatures. We propose that straight-chain fatty acid synthesis, reduced to sustain growth at lower temperature, is maintained due to inefficient turnover of the pyruvate dehydrogenase mRNA, leading to elevated acetyl-CoA levels. This study allowed us to unravel at least in part the cold sensitive phenotype and to show that the pyruvate dehydrogenase activity plays an important function in the regulation of fatty acid composition of the membrane, a process that remains poorly understood in Gram-positive bacteria.

## Introduction

Adaptation to different environments requires a sophisticated and complex gene expression system, allowing bacteria to quickly regulate mRNA abundance and thereby protein synthesis. Among the players involved in regulation of gene expression are DEAD-box RNA helicases, which are ubiquitous proteins found in all organisms [1,2]. DEAD-box RNA helicases display RNA-dependent ATPase and ATP-dependent RNA helicase activities that generally mediate local unwinding of an RNA duplex but can also achieve other molecular functions such as annealing, protein displacement, or clamping on RNA. Thus DEAD-box helicases act at various RNA remodeling steps that may be crucial for gene expression or ribonucleoprotein complex (RNP) assembly. In bacteria the functions established for DEAD-box RNA helicases are limited to translation initiation, ribosome biogenesis and RNA decay [1]. *Staphylococcus aureus* encodes two DEAD-box RNA helicases, CshA and CshB. In *Bacillus subtilis*, the two homologues have been proposed to be involved in ribosome biogenesis, however their functions in this process have not been detailed [3]. The discovery, in both *B. subtilis* and *S. aureus*, that CshA interacts with exoribonucleases such as RNase Y, RNase J1/J2 or PNPase that were proposed to form a degradosome, suggests that it is involved in mRNA turnover [4–6]. In *S. aureus* inactivation of *cshA* was first described in a screen aimed to find mutations that affect biofilm formation [7]. We later showed that this defect correlates with the requirement of CshA to efficiently degrade the *agr* mRNA that encodes the major quorum sensing system of *S. aureus*, governing many virulence factors [8]. Global scale mRNA half-life measurements showed that absence of CshA slightly increases the stability of the bulk mRNA and more significantly a subset of about hundred genes, corroborating a general function of CshA in mRNA decay [9]. In addition to these functions, an RNA protective role of CshA has also been proposed where the helicase protects some mRNAs upon MazE toxin induction [10], but this has not been studied further. Altogether these data suggest that the CshA DEAD-box RNA helicase might have various functions in RNA metabolism.

Inactivation of *cshA* also leads to a cold sensitive phenotype with a marked growth delay at 25°C. Cold sensitivity is a common phenotype shared by inactivation of various RNA

helicases. It is believed that, as temperature is reduced, the stronger stability of RNA duplexes increases the dependency on RNA helicase. Inactivation of CshB, the second DEAD-box protein of *S. aureus* also altered cold–growth. This phenotype has been related to a magnesium homeostasis defect, although the molecular details have not been worked out [11]. In *E. coli*, the cold sensitivity of *csdA* and *srmB* mutants have been related to defects in ribosome biogenesis [12,13].

In order to understand the reason of the cold sensitivity in absence of CshA, we performed a large genetic screen selecting spontaneous suppressors that improve the growth of the helicase mutant strain at 25˚C. The identification of various mutants affecting the fatty acid biosynthesis system (FASII) clearly identified membrane biosynthesis as one of the impaired pathways. Changes of the fatty acids composition is known as a major mechanism in response to low temperature in order to counteract rigidification of the membrane by introducing lower melting-point molecules [14,15]. In Gram-negative bacteria this is achieved by increasing the synthesis of unsaturated fatty acid (UFA). Membranes of Gram-positive bacteria are majorly composed of branched-chain fatty acids (BCFA) and straight-chain fatty acids (SCFA). Response to low temperature in these organisms relies essentially on the increase of BCFA production associated with a decrease in chain length by an undefined mechanism [16]. Here we show that this cold-induced response is impeded in the *ΔcshA* strain explaining at least in part its cold sensitivity. Thus our data link for the first time the cold sensitivity of a DEAD-box RNA helicase mutant to a membrane biosynthesis defect. Deeper molecular analysis allowed us to propose that the inability of the mutant to increase BCFA content is due to the requirement of CshA for the degradation of the mRNA of the pyruvate dehydrogenase (PDH) operon, which synthetizes acetyl-CoA [17]. BCFA are derived from branched-chain amino acids, whereas SCFA are derived from acetyl-CoA. We propose that the accumulation of the *pdh* mRNA in the *ΔcshA* strain at low temperature exacerbates the production of SCFA, at the expense of BCFA production.

In addition to showing the importance of mRNA decay pathways and in particular the function of the CshA DEAD-box protein in membrane homeostasis, this work also emphasizes the importance on the connection between central metabolism and fatty acids homeostasis.

## Results

### Mutations in genes involved in fatty acids metabolism suppress cold-sensitivity of *ΔcshA*

To get insight into the biological pathways impeded in a *ΔcshA* mutant we performed a genetic screen to isolate spontaneous suppressors that improve growth at 25˚C. At this temperature the *ΔcshA* mutant did not cause the death of the bacteria but showed a pronounced growth delay with colonies appearing in 4 to 5 days compared to the *wt* strain that formed colonies in less than 2 days. We selected 82 suppressor strains by plating independent over-night cultures of a *ΔcshA* strain on Mueller Hinton broth (MH) plates at 25˚C. Note that the suppressor strains showed only partial restoration of the growth with colonies appearing after 3 days of incubation. All selected strains were subjected to whole genome sequencing using a strategy where genomic DNAs were pooled in 9 groups before library preparation. Using an in-house *de novo* assembly-based software, SNPs and INDELs as well as larger rearrangements or transposition events were identified (see materials and methods). All the mutations used in this study were further confirmed by Sanger-sequencing and assigned to a specific suppressor strain (S1 Table). This primary analysis enabled us to attribute the number of potential mutations of interest to 16 genes (Table 1). A major outcome of the genetic screen is the high occurrence of mutations in genes linked to membrane synthesis (Fig 1, Table 1). Among them, we

identified mutations in genes involved in fatty acid metabolism. In particular, *accC* and *accD* genes encode two subunits of the acetyl-CoA carboxylase (ACC complex) catalyzing the first step of fatty acid synthesis; *fabD* gene that encodes the malonyl-CoA-acyl carrier protein transacyclase responsible for the second step of fatty acid synthesis (Fig 1). Multiple mutations were also identified in *fakA*, encoding a fatty acids kinase [18], or in *ahrC*, encoding a transcriptional regulator shown latter here to direct the synthesis of the Branched-Chain Keto-acid Dehydrogenase (BCKD) complex involved in the synthesis of the BCFA precursors [19,20]. These results suggested unambiguously that suppression of the cold sensitive phenotype of the *ΔcshA* strain mainly goes through modulation of the membrane composition. We therefore set out to analyze the effect of some of the mutations on the synthesis of fatty acid to understand the basis of suppression.

## Decreased activity of FASII suppresses the *ΔcshA* cold-growth phenotype

The formation of malonyl-CoA and its interaction with FapR (fatty acid and phospholipid regulator), a transcriptional repressor of 4 operons, was shown to be a key point in the fatty acid and phospholipid synthesis pathway [21–23]. FapR-mediated transcriptional control of genes encoding FASII and phospholipid biosynthesis proteins is, so far, the only well-characterized regulation of these processes established in *S. aureus*. This feedforward mechanism enables the expression of its targets only if the substrate is available and links the expression of genes involved in both fatty acid and phospholipid synthesis. Interestingly, 5 mutations were located in 2 genes of the 4 subunits of the ACC complex, AccC and AccD, which catalyzes the first step of fatty acid synthesis producing malonyl-CoA from acetyl-CoA (Table 1 and Fig 1). In order to assess whether these mutations were increasing or decreasing the activity of the complex, we monitored the repressor function of the malonyl-CoA sensitive FapR. If malonyl-CoA is elevated and binds to FapR, the repressor cannot bind to the operator and synthesis occurs,

**Table 1. Mutations retrieved in the genetic screen.**

| gene function | number of mutations | mutations |
|---|---|---|
| Acetyl-coenzyme A carboxylase | 5 | $accC^{M385V}$, $accC^{T183I}$ |
| | | $accD^{A164E}$, $accD^{A164V}$, $accD^{F253V}$ |
| Arginine repressor | 2 | $ahrC^{N31Stop}$, $ahrC^{R44C}$ |
| biotin pathways | 3 | $bioY^{P123R-FsX1}$ |
| | | $birA^{D320F-FsX28}$, $birA^{R280stop}$ |
| Malonyl CoA-acyl-carrier-protein transacylase | 1 | $fabD^{Q164L}$ |
| Fatty acid kinase | 35 | $fakA^{(1)}$ |
| Lipoprotein signal peptidase | 1 | $lspA^{R51H-FsX11}$ |
| Lipoteichoic acid synthase | 12 | $ltas^{(1)}$ |
| NADH dehydrogenase | 5 | $ndhf^{Gdel\ -8nt}$, $ndhF^{ins\_IS1181\ -12nt}$, $ndh^{FA223P}$, $ndhF^{S263F}$, $ndhF^{V267E}$ |
| similar to nitrogen fixation protein | 1 | $nfu^{G>A-13nt}$ |
| Pyruvate dehydrogenase | 5 | $pdhA^{A2P}$, $pdhA^{E362Stop}$, $pdhA^{G126D}$ |
| | | $pdhB^{P118L}$ |
| | | $pdhC^{D42Y}$ |
| Unknown, in operon with *fakA* | 1 | SA1068$^{G35V}$ |
| δ-subunit of RNA polymerase | 1 | $rpoE^{S83\_D86dup}$ |

[1]see S1 Table for all mutations

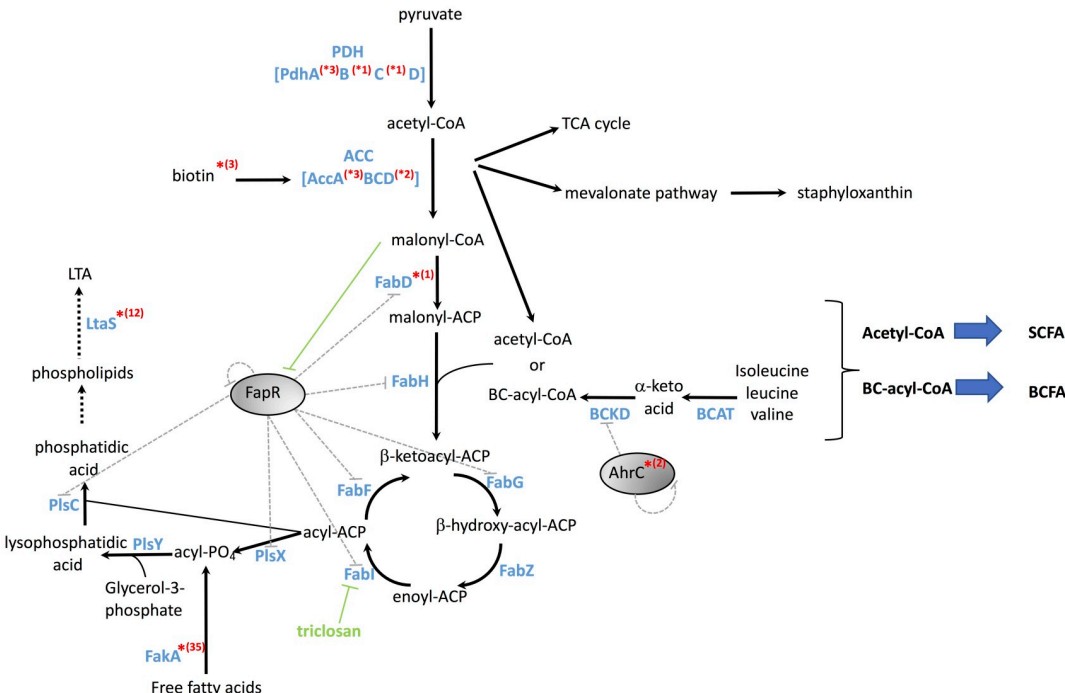

**Fig 1. Schematic of fatty acid and phospholipid synthesis pathways.** Presence of mutations of the genetic screen at the various steps are indicated by a red asterisk, with the number of mutations indicated in parenthesis. Genes targeted by the two transcriptional repressors FapR and AhrC are indicated with grey dotted arrows. Repression of FapR by malonyl-CoA and inhibition of FabI by triclosan are indicated with green arrows. ACC: acetyl-coA carboxylase complex; FabD: malonyl-CoA-acyl-carrier-protein transacyclase; FabH: Beta-ketoacyl-acyl-carrier-protein synthase III; FabG: 3-oxoacyl-acyl-carrier-protein reductase; FabZ: 3-hydroxyacyl-acyl-carrier-protein dehydratase; FabI: Enoyl-acyl-carrier-protein reductase; FabF: 3-oxoacyl-acyl-carrier-protein synthase 2; BCAT: Branched-chain aminotransferase; BCKD: Branched-Chain α-keto acid dehydrogenase complex; BCFA: Branched-Chain Fatty Acids; SCFA: Straight-Chain Fatty Acids; TCA cycle: tricarboxylic acid cycle. PlsX: phosphate acyl transferase; PlsY: glycerol-3-phosphate acyltransferase; PlsC: 1-acyl-sn-glycerol-3-phosphate acyl transferase; FakA: fatty acid kinase; LTA: LipoTeichoic Acid, LtaS: LipoTeichoic Acid Synthase; FapR: fatty acid and phospholipid regulator; AhrC: arginine metabolism transcriptional regulator.

whereas if the ACC activity is low, resulting in low malonyl-CoA concentration, the repressor binds to its targets. We therefore measured the *fapR* messenger level, which is part of the *fapR-plsX-fabD-fabG* operon and controlled by itself, by RT-qPCR in exponentially growing cells cultivated at 25˚C. The results show first that *fapR* operon expression is increased 4-fold in *ΔcshA* compared to the *wt* strain (discussed later) and second and importantly that in the 4 ACC suppressor strains tested the *fapR* operon expression decreased by about 2-fold compared to the parental *ΔcshA* strain, suggesting a lower production of malonyl-CoA (Fig 2A).

Moreover, we identified three mutations located in genes involved in biotin synthesis or uptake pathways (Table 1), with two mutations in *birA*, encoding the bifunctional biotin ligase and transcriptional repressor, and one in *bioY*, the substrate-specific S component of the biotin ECF transporter [24]. Since biotin is a cofactor of the ACC complex, we determined whether its activity is also affected by these mutations using the same read-out than above. Fig 2A shows that indeed the level of *fapR* mRNA decreased in the 3 biotin-related mutants when compared to the *ΔcshA* strain, suggesting a loss of ACC activity in these strains. We also obtained a mutation in the gene encoding FabD (*fabD*$^{Q164L}$), which performs the second step of fatty acid biosynthesis (Fig 1), i.e. the production of malonyl-ACP from malonyl-CoA. However, this suppressor strain also contains another mutation in the *pknB* gene. To ascertain

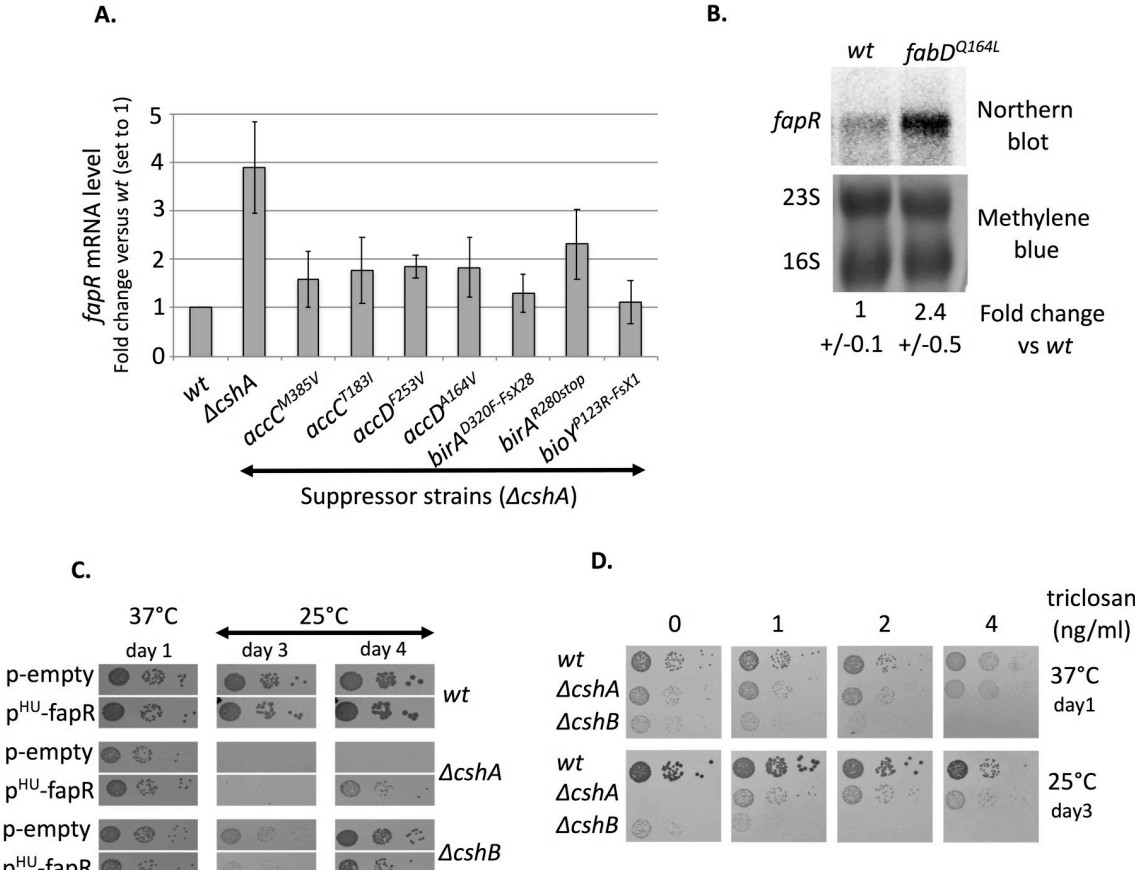

**Fig 2. Decreased activity of FASII favors the cold growth of the Δ*cshA* strain.** (**A**) *fapR* mRNA levels decrease in ACC and biotin related suppressor strains. *fapR* mRNA levels were quantified by RT-qPCR using 16S rRNA as reference gene on total RNAs extracted from *wt* strain (PR01), Δ*cshA* strain (PR01-Δ*cshA*), and suppressor strains Δ*cshA/accC*$^{M385V}$ (C51), Δ*cshA/accC*$^{T183I}$ (sup30), Δ*cshA/accD*$^{F253V}$ (sup17), Δ*cshA/accD*$^{A164V}$ (sup16), Δ*cshA/birA*$^{D320F-FsX28}$ (C58), Δ*cshA/birA*$^{R280stop}$ (sup1) and Δ*cshA/bioY*$^{P123R-FsX1}$ (C66), all at exponential growth phase in MH medium at 25˚C. n = 5 for Δ*cshA*, 4 for *wt* and 3 for all others. Expression is shown as arbitrary units, where the *wt* level is set to 1. Standard deviations are represented. Numerical data are shown in S1 File. (**B**) *fapR* mRNA level increases in a *fabD*$^{Q164L}$ mutant. Total RNA was extracted from exponentially growing cells in MH medium at 37˚C in *wt* (PR01) and *fabD*$^{Q164L}$ (SVK86). Top panel: Northern blot of RNA separated by agarose/formaldehyde gel electrophoresis, hybridized with a probe against the *fapR* gene. Lower panel: methylene blue coloration of the membrane used in top panel; 16S and 23S rRNA are indicated. Expression level of *fapR* mRNA from phosphoimager quantification is shown as arbitrary units, where *wt* level is set to 1, and is the result from 4 biological replicates for each strain. Standard deviations are indicated. (**C**) Over-night cultures of transformants of *wt* (PR01), Δ*cshA* (PR01-Δ*cshA*) and Δ*cshB* (PR01-09) strains, carrying either the empty vector (pCN47) or p$^{HU}$-fapR (pVK102), a plasmid expressing FapR under control of the constitutive *hu* promoter were serially diluted and spotted on MH plates containing erythromycin, and incubated at 37˚C or 25˚C. (**D**) Addition of triclosan favors Δ*cshA* growth at 25˚C. Over-night cultures of *wt* (PR01), Δ*cshA* (PR01-Δ*cshA*) and Δ*cshB* (PR01-09) strains, were serially diluted and spotted on MH plates containing uracil and triclosan at the indicated concentrations and incubated at 37˚C or 25˚C.

that the *fabD* mutation was a true suppressor of Δ*cshA*, both the *fabD* and the *pknB* mutations were reconstructed in Δ*cshA*. The results show that the mutation in *fabD* indeed suppresses the cold sensitivity of Δ*cshA*, whereas the *pknB* mutation does not (S1 Fig). When FapR regulon expression was measured in a *fabD*$^{Q164L}$ mutant, in an otherwise *wt* background, we observed an accumulation of *fapR* mRNA (Fig 2B). This is in agreement with a decreased activity of FabD, leading to an accumulation of the unused malonyl-CoA and therefore higher expression of FapR regulon.

The above results indicate that lowering down FASII activity favors the growth of a Δ*cshA* strain at 25˚C. To confirm this result, we first attempted to decrease expression of the overall

FapR regulon by over-expression of the FapR repressor. For this purpose *fapR* was cloned on a multicopy plasmid under the control of the *hu* constitutive promoter leading to the p$^{HU}$-fapR plasmid. Fig 2C shows that the presence of p$^{HU}$-fapR in Δ*cshA* was able to improve its growth at 25˚C. This effect was specific to the Δ*cshA* strain as introduction of p$^{HU}$-fapR in both *wt* or Δ*cshB* strain, another cold-sensitive DEAD box RNA helicase mutant, does not improve growth (Fig 2C). Second, we used the disinfectant triclosan that targets the enoyl reductase FabI protein of the FASII cycle (Fig 1). Interestingly, the results showed that addition of triclosan, at sub-lethal doses, also improved the growth of the Δ*cshA* strain at 25˚C, whereas the growth of Δ*cshB* was not restored (Fig 2D). The above results establish a clear link between the activity of FASII and the inability of Δ*cshA* strain to grow in the cold.

## Increased BCKD synthesis favors growth of a Δ*cshA* strain at low temperature

While the mutations described above clearly link the suppression of the cold sensitivity of a Δ*cshA* strain with decreased activity of the FASII system, it does not explain by itself why. Mutations in the *ahrC* gene reveal an interesting possible answer. The *ahrC* gene, mutated independently twice in the suppressor screen, is a transcriptional repressor of the arginine biosynthesis pathway (Table 1) [25]. One mutation, *ahrC*$^{R44C}$, is located in its DNA binding motif whereas the second, *ahrC*$^{N31Stop}$, introduces a stop codon at the very beginning of the ORF indicating that inactivation of *ahrC* favors the growth of Δ*cshA* in the cold. Whereas the arginine biosynthesis pathway and FASII activity have not been interconnected so far, the genetic organization of *ahrC* found in *S. aureus* offers a possible explanation that links these mutations to membrane synthesis. Indeed, *ahrC* is found upstream of the *recN* gene and the 4 genes (*lpdA*, *bfmAAA*, *bfmBAB*, *bfmBB*) encoding the BCKD complex involved in the synthesis of branched-chain fatty acid (BCFA) precursors (Fig 3A) [19,20]. As AhrC regulates its own gene, we hypothesized that its inactivation leads to an increase of the expression of the BCKD genes. We therefore measured the *bfmBAA* mRNA level by RT-qPCR in the 2 *ahrC* suppressor strains and compared it to Δ*cshA* grown at mid-exponential phase at 25˚C. The results showed that indeed *bfmBAA* expression increased more than 2-fold in the two *ahrC* mutants (Fig 3B). Interestingly, we showed that, whereas the expression of wt *ahrC* from a multicopy plasmid eliminated the effect of both suppressor mutations, the mutant *ahrC*$^{R44C}$ on a plasmid in a Δ*cshA* strain suppressed the cold sensitivity indicating a dominant negative effect of the mutation (Fig 3C). Moreover, the presence of wt *ahrC* on a multicopy plasmid in a *wt* strain resulted in a cold sensitive growth defect (Fig 3C), consistent with reduced expression of the BCKD enzymes [20]. However, we could not distinguish between an effect due to decreased expression of the AhrC regulon or an effect of BCKD gene down-regulation. We therefore expressed the BCKD genes on a multicopy plasmid and showed that, in accordance with our hypothesis, this over-expression improved specifically the growth of the Δ*cshA* strain at low temperature (Fig 3D), whereas the Δ*cshB* cold growth was unchanged. The above results indicate that BCKD synthesis is driven at least in part by the AhrC-controlled promoter and suggests that BCKD over-expression and hence increased BCFA production favors growth of the helicase mutant at low temperature.

## The RNA helicase mutant can no longer increase branched-chain fatty acids

Fatty acids composition of the membrane is highly changing depending on strains, media, and especially on the temperature with an increased BCFA production associated with a decrease of chain length, required to sustain growth at lower temperature [14,15]. As the above results

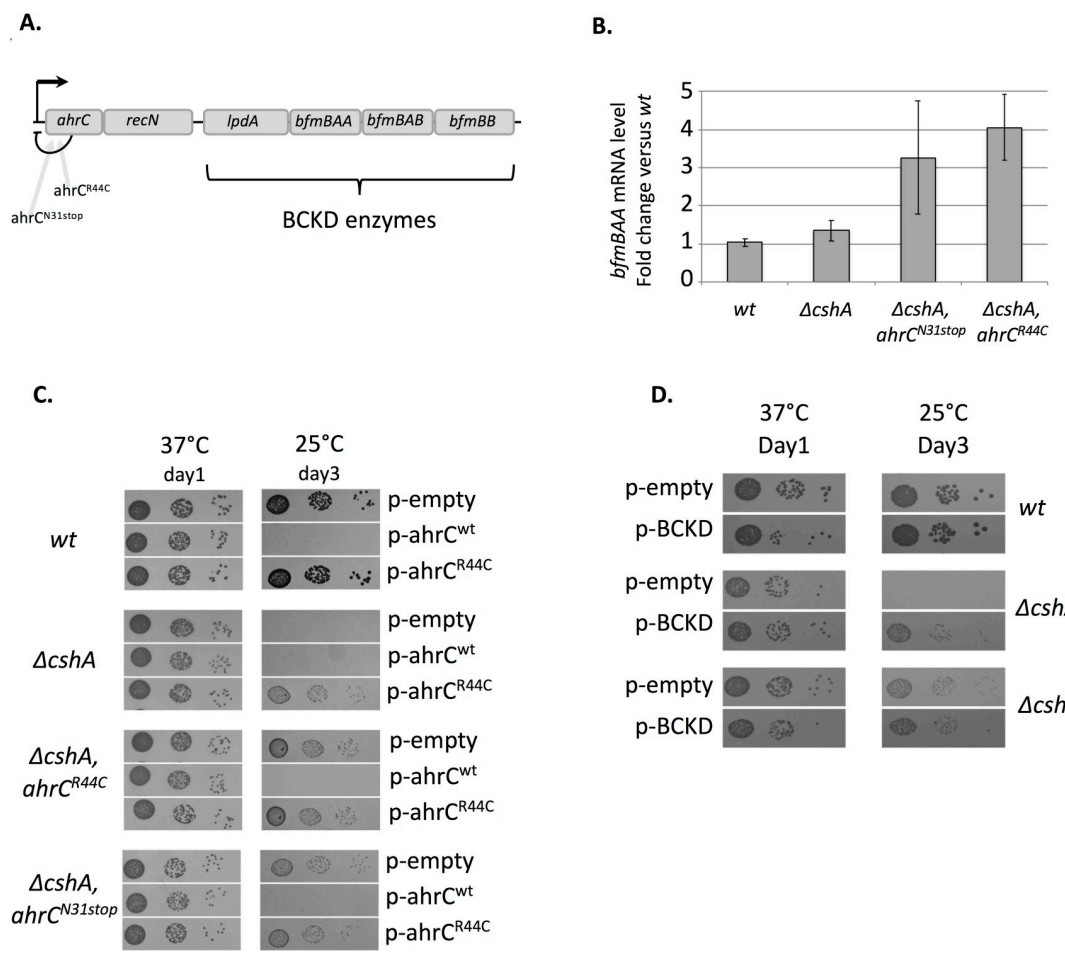

**Fig 3. BCKD over-expression favors Δ*cshA* cold growth. (A)** Genetic organisation of the *ahrC* locus. Schematic representation of the genomic locus of the transcriptional repressor *ahrC* (not at scale). The mutations in *ahrC* obtained in the screen are indicated. *lpdA*, *bfmBAA*, *bfmBAB*, and *bfmBB* encode proteins of the Branched-Chain α-keto acid dehydrogenase complex (BCKD). **(B)** *bfmBAA* mRNA level increase in *ahrC* mutants. Total RNA was extracted from exponentially growing cells in MH medium at 25˚C, in *wt* strain (PR01), Δ*cshA* strain (PR01-Δ*cshA*), and suppressor strains Δ*cshA/ahrC^N31stop^* (C9) and Δ*cshA/accD^R44C^*(C57). *bfmBAA* mRNA levels were quantified by RT-qPCR using 16S rRNA as reference gene. Expression is shown as arbitrary units, where the *wt* level is set to 1. Standard deviations are represented. n = 4 for *wt* and Δ*cshA* and 3 for the others. Numerical data are shown in S1 File. **(C)** Effect of over-expression of wt allele and mutant allele of AhrC in *wt*, Δ*cshA* and *ahrC* suppressor strains. Over-night cultures of transformants of strains *wt* (PR01), Δ*cshA* (PR01-Δ*cshA*), Δ*cshA/ahrC^R44C^* (C57) and Δ*cshA/ahrC^N31stop^* (C9) with the empty vector (pCN47) or plasmids expressing AhrC *wt* (pVK174) or AhrC^R44C^ (pVK176), were serially diluted and spotted on MH plates containing erythromycin, and incubated at the indicated temperatures. **(D)** Expression of BCKD complex from a multicopy plasmid partially restores Δ*cshA* growth at 25˚C. Over-night cultures of transformants of strains *wt* (PR01), Δ*cshA* (PR01-Δ*cshA*) and Δ*cshB* (PR01-09) with empty vector (pCN47) or a plasmid expressing the four BCKD genes (pVK197) were serially diluted and spotted on MH plates containing erythromycin, and incubated at the indicated temperatures.

indicate a requirement for over-expression of the BCKD complex to favor Δ*cshA* cold growth, we analyzed the fatty acid profile in *wt* and Δ*cshA* strains to see whether membrane accommodation was effective. Interestingly, we observed during our study that the suppressor mutations efficiently favored the growth of a Δ*cshA* when plated at 25˚C, whereas in liquid they did not improve growth of the helicase mutant, and in some cases made it even worse. The suppressors are thus specific for the screening condition and we therefore analyzed the fatty acid content from cells harvested from solid medium. At 37˚C in *wt* SCFA represent only 17.4%, whereas BCFA are highly abundant reaching 82.6% in Muller Hinton, which has already been reported

in[26](Fig 4A). When lowering the temperature, differences are therefore subtle but reproducible and SCFA decreased as expected from 17.4% at 37˚C to 13.9% at 25˚C in the *wt* strain (Fig 4A), and chain length decreases (S2 File). Remarkably and in contrast to the *wt* strain, the SCFA content in *ΔcshA* mutant stayed constant when changing from 37˚C and 25˚C, remaining at about 20% at 25˚C, while chain lengths are overall longer than in the *wt* strain (S2 File). As a control, we analyzed the fatty acid composition of *ΔcshB* strain, which is also cold sensitive, and showed that it is similar to the *wt* strain decreasing from 18.1% at 37˚C to 14.2% at 25˚C, indicating that fatty acid composition in *ΔcshA* is specific and not a consequence of slow growth. We next analyzed the fatty acid profile in a set of suppressors incubated at 25˚C and found that in all cases the BCFA content was restored close to or even slightly higher than in the *wt* (Fig 4B). This result suggests that the *ΔcshA* strain is not able to accommodate its membrane at low temperatures, whereas the selected suppressor strains restore this defect.

## Increased *fapR* operon mRNA level does not explain the cold sensitivity of a *ΔcshA* strain

As mentioned above, the *fapR* operon, which comprises *fapR* but also *plsX*, *fabG* and *fabD*, is over-expressed about 4-fold at 25˚C in a *ΔcshA* strain. Such an increase is in agreement with our previous study on the global effect of the absence of CshA on mRNA decay, which showed that *fapR* mRNA half-life increased about 2-fold in a *ΔcshA* at 37˚C [9]. In *B. subtilis* the deletion of the *fapR* gene, which leads to over-expression of the entire FapR regulon, results in a cold sensitive phenotype associated with fatty acids changes [23]. We therefore investigated whether the *fapR* operon mRNA accumulation in *ΔcshA* could explain its cold sensitivity. However and in sharp contrast with *B. subtilis*, the deletion of *fapR* does not confer any cold sensitivity even if the temperature is lowered down to 16˚C (Fig 5A). We confirmed that two targets of FapR, the *fapR-plsX-fabD-fabG* and the *fabH-fabF* operons were overproduced in the *ΔfapR* strain (Fig 5B). Note that the *fapR-plsX-fabD-fabG* operon accumulated in *ΔfapR* to a similar extent than in *ΔcshA*. These observations tend by themselves to invalidate the hypothesis of a possible effect of the *fapR* operon accumulation. Nonetheless, we introduced the *fapR* deletion in two *acc* and one *pdh* (pyruvate dehydrogenase complex, described below) suppressor strains. In the 3 cases the suppressor mutations were still able to favor the *ΔcshA* at lower temperature (Fig 5C), indicating that over-expression of FapR targets does not severely impact cold growth in *S. aureus*.

## Over-expression of pyruvate dehydrogenase as key to explain the cold sensitivity

In our suppressor screen we identified five mutations in 3 of the 4 genes encoding the subunits of the pyruvate dehydrogenase complex (PDH) responsible for the conversion of pyruvate to acetyl-CoA (Table 1, Fig 1). Acetyl-CoA is at the cross-road of many metabolic pathways and is used in particular at two different steps in fatty acid synthesis (Fig 1). First it is used by ACC to produce malonyl-CoA [27], and second by FabH, which uses either acetyl-CoA or branched-chain fatty acid precursors to produce SCFA or BCFA, respectively [28]. To determine whether the *pdh* mutants affect ACC activity, we measured as above *fapR* mRNA level in 3 *pdh* mutants grown on plates at 25˚C. The results show that *fapR* mRNA level stays unchanged in the *pdh* suppressor strains tested when compared to *ΔcshA*, which suggests that the mutations do not have a major impact on ACC activity (Fig 6A). We thus hypothesized that *pdh* mutants affect FabH activity, which fits with a recent study that showed an increased BCFA production when PDH was inactivated [29]. In accordance with this hypothesis, Fig 4B shows that *pdh* mutations lead to an important increase of BCFA in *ΔcshA* at 25˚C. We next

**A.**

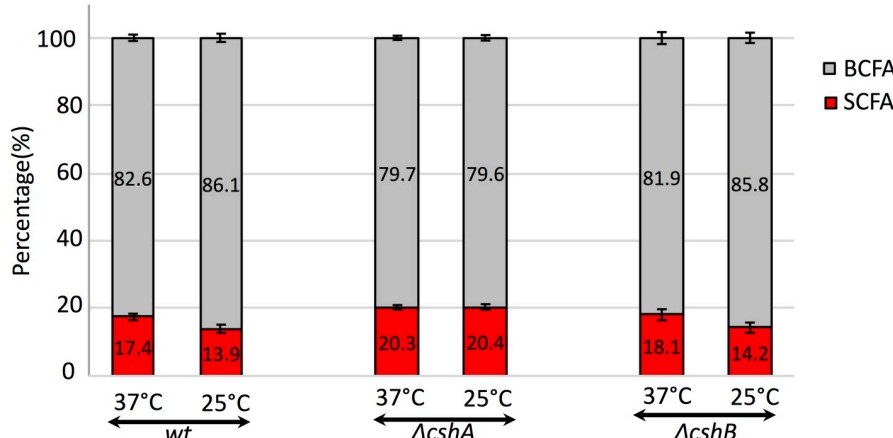

**B.**

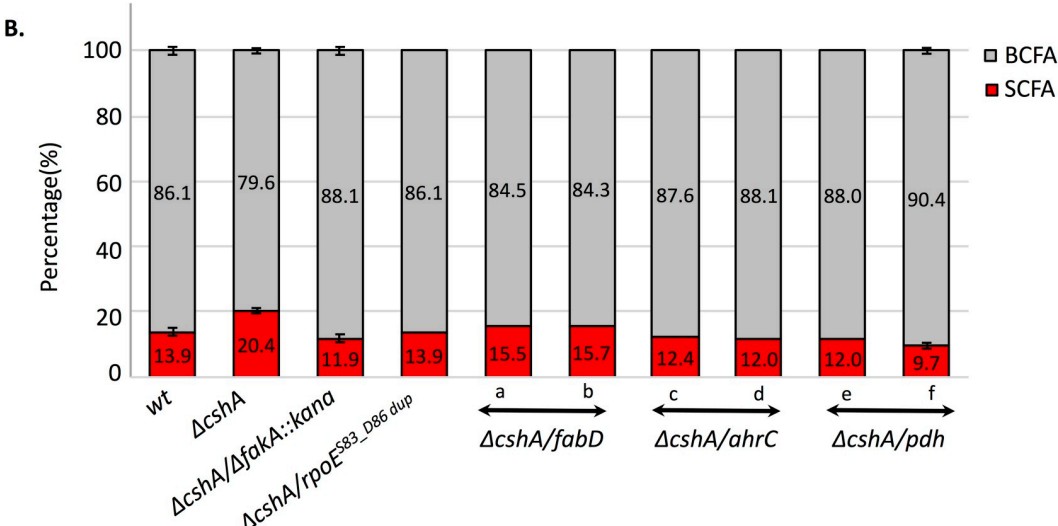

**Fig 4. *ΔcshA* strain fails to accommodate its membrane to low temperature. (A)** Changes of fatty acid profile at 37°C versus 25°C. Relative percentage of BCFA and SCFA measured by gas chromatography from *wt*, *ΔcshA* and *ΔcshB* strains scrapped from agar plates incubated at 37 or 25°C. Mean and standards deviation are represented (n = 3 for *wt* and *ΔcshA* and 2 for *ΔcshB*). See complete data set in S2 File **(B)** BCFA content is restored in suppressor strains. Relative percentage of BCFA and SCFA measured by gas chromatography from *wt*, *ΔcshA* and suppressor strains scrapped from agar plates incubated at 25°C. Mean and standard deviations are represented (n = 3 for *wt* and *ΔcshA*, 2 for *ΔcshA/ΔfakA::kana* (SVK48) and *ΔcshA/pdh*$^{A2P}$ (f: C43). *fabD*$^{Q164L}$ cumulated with *ΔcshA*, was tested twice both with the selected suppressor strain (a: C53) or the reconstructed one (b: SVK92). The two suppressor strains containing mutations in *ahrC* were both tested once: c = *ΔcshA/ahrC*$^{N31stop}$ (C9) and d = *ΔcshA/ahrC*$^{R44C}$ (C57). e is a second mutation in *pdh*, tested once: *ΔcshA/pdhA*$^{E362stop}$ (sup26). See S2 File for complete data set and analysis by fatty acid types and chain length.

analyzed whether this was specific to the tested condition or also occurring at 37°C and in a *wt* background. The results in Fig 6B show that *pdh*$^{A2P}$ mutation has a large impact on fatty acids composition in all tested conditions, with the smallest amount of SCFA (6.3%) measured all along this study when *pdh*$^{A2P}$ was grown at 25°C. Interestingly we noticed previously that the *pdh* mRNA decayed slightly slower in a *ΔcshA* mutant at 37°C [9] and therefore we determined the *pdh* mRNA half-life in both *wt* and *ΔcshA* grown at 25°C and found that the *pdh*

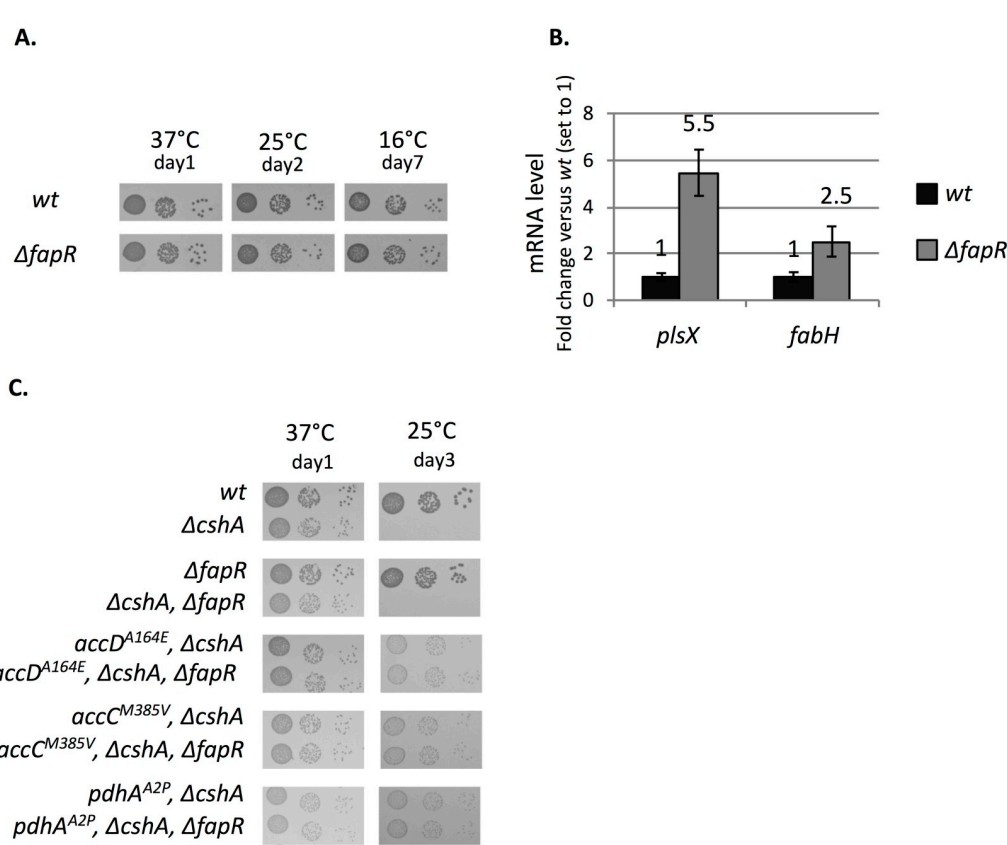

**Fig 5. *fapR* operon over-expression does not explain the *ΔcshA* cold sensitivity. (A)** Deletion of *fapR* does not affect growth. Over-night cultures of *wt* (PR01, top panels) and *ΔfapR* (SVK31, bottom panels), were serially diluted and spotted on MH plates and incubated at the indicated temperatures. **(B)** Total RNA were extracted from exponentially growing cells in MH medium at 25˚C, in *wt* strain (PR01) and *ΔfapR* strain (SVK31). *plsX and fabH* mRNA levels were quantified by RT-qPCR using 16S rRNA as reference gene. Expression is shown as arbitrary units, where the *wt* level is set to 1. Standard deviations are represented. n = 2 for *wt* and *ΔfapR*. Numerical data are shown in S1 File. **(C)** Deletion of *fapR* does not interfere with the suppressor phenotype of *acc* or *pdh* mutants. Over-night cultures of *wt* (PR01), *ΔcshA* (PR01-ΔcshA), *ΔfapR* (SVK31), *ΔcshA/ΔfapR* (SVK32), *ΔcshA/accC^{M385V}* (C51), and its *ΔfapR* derived strain (C51-ΔfapR), *ΔcshA/ AccD^{A164E}* (C1), its *ΔfapR* derived strain (C1-ΔfapR), the *ΔcshA/pdhA^{A2P}* (C43), and its *ΔfapR* derived strain (C43-ΔfapR) were serially diluted and spotted on MH plates and incubated at 37˚C (left panels) or 25˚C (right panels).

mRNA in *ΔcshA* was further stabilized, above 2-fold, possibly because the RNA structure(s) that require CshA is strengthened at lower temperature (Fig 6C). We then measured the steady state level of *pdh* mRNA in the specific conditions of this study (ie., on plates incubated at 25˚C) and showed that indeed the entire polycistronic *pdh* mRNA accumulated in *ΔcshA* compared to the *wt* strain (Fig 6D). We therefore tested whether *pdh* over-expression by itself would interfere with growth at reduced temperature, by expression of the entire operon on a multicopy plasmid. The results show that in presence of the *pdh* operon on a plasmid, the growth was reduced at 37˚C and severely impacted at 25˚C (Fig 6E, upper panel). As acetyl-CoA is used by many processes, the observed effect could be due to erroneous regulation of various pathways and does not show by itself that *pdh* over-expression was related to imbalance of BCFA/SCFA ratio. We therefore introduced the p-PDH plasmid into the *fabD* mutant identified in the *ΔcshA* suppressor screen and reconstructed in a *wt* background, and found that its effect on growth was much less severe (Fig 6E, bottom panels). Moreover, addition of 2 ng/ml of triclosan also favored the growth of the PDH over-expressing strain (Fig 6E, middle

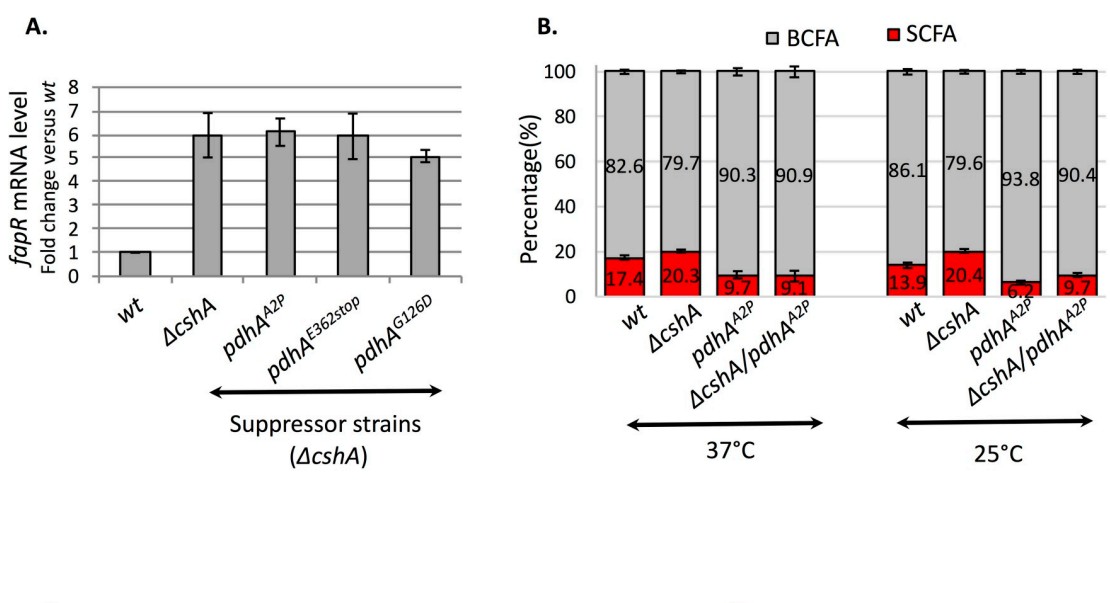

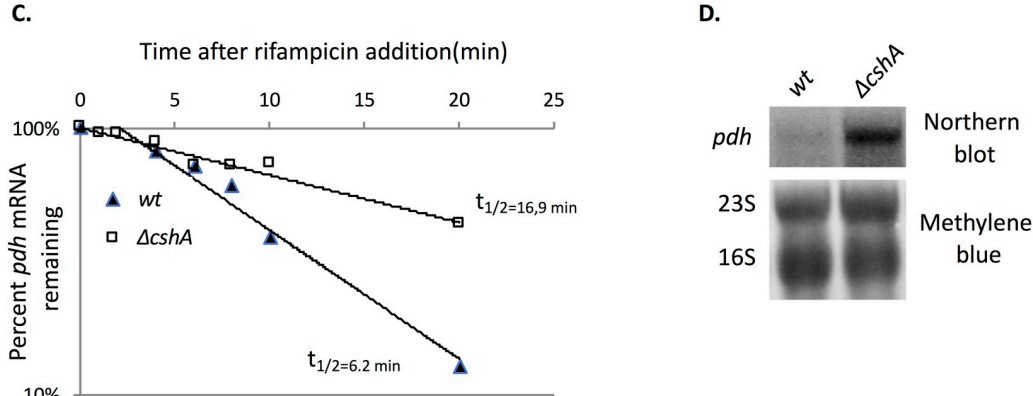

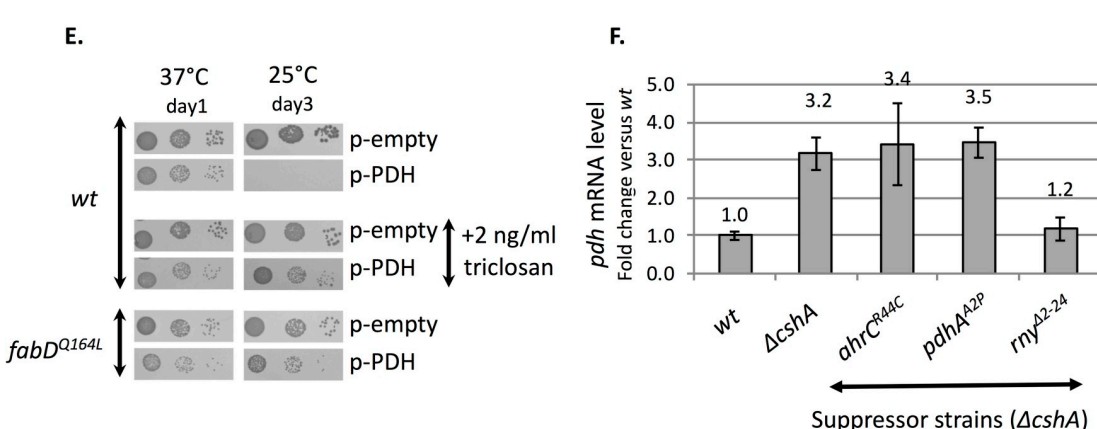

**Fig 6. PDH over-expression in *ΔcshA* may explain its cold sensitivity. (A)** *fapR* expression is not influenced by *pdh* mutations. Total RNA was extracted from cells scrapped from MH agar plates containing uracil and incubated at 25˚C. The strains were *wt* (PR01), *ΔcshA* (PR01-ΔcshA), *ΔcshA/pdhA^{A2P}* (C43), *ΔcshA/pdhA^{E362stop}* (sup26), *ΔcshA/pdhC^{D42Y}* (C12), *ΔcshA/pdhA^{G126D}* (C45) and *ΔcshA/pdhB^{P118L}* (C55). *fapR* mRNA levels were quantified by RT-qPCR using 16S rRNA as reference gene. Expression is shown in arbitrary units, where the wt level is set to 1. Standard deviations are represented for PR01 (n = 4), PR01-ΔcshA (n = 3),

C43 (n = 5), sup26 (n = 5) and C45(n = 3). Numerical data are shown in S1 File. **(B)** Increased BCFA content in *pdh* mutants. Relative percentage of BCFA (iso-odd, iso-even and anteiso) and SCFA from *wt*, *ΔcshA*, *pdh*$^{A2P}$ (SVK88) and *ΔcshA/pdh*$^{A2P}$ (C43) strains scrapped from plates incubated at 37 or 25°C. Mean and standard deviations are represented (n = 3 for *wt* and *ΔcshA* and 2 for *pdh*$^{A2P}$ and *ΔcshA/pdh*$^{A2P}$). See complete data set in S2 File. **(C)** *pdh* mRNA half-life increase in *ΔcshA*. Total RNA was extracted from *wt* (PR01) and *ΔcshA* (PR01-ΔcshA) from exponentially growing cells in MH at 25°C, 0, 1, 2, 4, 6, 8, 10 and 20 min after rifampicin addition at 400 μg/ml. The mRNA levels of SA0944, the second gene of the *pdh* operon encoding PdhB, were quantified by RT-qPCR using 16S rRNA as reference. Expression is shown in arbitrary units, where time 0 after rifampicin addition was set to 100% and the decay plotted on a semi-logarithmic scale. The calculated half-life of *pdh* mRNA is indicated for each strain. Numerical data are shown in S1 File. **(D)** *pdh* mRNA accumulate in *ΔcshA* grown on plate at 25°C. Total RNA was extracted from *wt* (PR01) and *ΔcshA* (PR01-ΔcshA) cells scrapped from plates incubated at 25°C. Northern blot of RNA separated by formaldehyde/agarose gel electrophoresis and hybridized with a probe against the *pdhD* gene (top panel). Lower panel: methylene blue coloration of the membrane used in top panel; 16S and 23S rRNA are indicated. **(E)** Expression of the *pdh* operon from a multicopy plasmid severely impacts growth in the cold but is partially suppressed by a *fabD*$^{Q164L}$ mutation or addition of triclosan. Over-night cultures of transformants of *wt* (PR01) or *fabD*$^{Q164L}$ (SVK86) strains with empty vector (pCN47) or a plasmid expressing the entire *pdh* operon (pVK203) were serially diluted and spotted on MH plates containing erythromycin, and incubated at the indicated temperatures. Indicated plates contain 2 ng/ml of triclosan. **(F)** Increased *pdh* mRNA level in *ΔcshA* is not changed by *pdh* or *ahrC* mutants but is lowered down by the *rny*$^{A2−24}$ truncation. Total RNA was extracted from cells incubated at 25°C and scrapped from MH agar plates. The strains were *wt* (PR01), *ΔcshA* (PR01-ΔcshA), *ΔcshA/pdhA*$^{A2P}$(C43), *ΔcshA/ahrC*$^{R44C}$ (C57) and *ΔcshA/rny*$^{A2−24}$ (SVK7). The mRNA levels of SA0944, the second gene of the *pdh* operon encoding PdhB, were quantified by RT-qPCR using 16S rRNA as reference gene. Expression is shown as arbitrary units, where *wt* was set to 1. Standard deviations are represented for *wt* (PR01, n = 6), *ΔcshA* (PR01-ΔcshA, n = 5), *ΔcshA/pdhA*$^{A2P}$ (C43, n = 3), *ΔcshA/ahrC*$^{R44C}$ (C57, n = 3) and *ΔcshA/rny*$^{A2−24}$ (SVK7, n = 4). Numerical data are shown in supplementary 2.

panels). It is to note that in both cases the suppressor effect is observed at 25°C, whereas the growth defect at 37°C was neither suppressed by the *fabD* mutation nor the addition of triclosan, suggesting that defect at 37°C may not be related to fatty acid synthesis and may be due to deregulation of another pathway using acetyl-CoA. These results showed that lowering the FASII activity compensates the over-expression of *pdh* as it was observed in the *ΔcshA* strain. This suggests that the growth defect in absence of CshA could be due, at least in part, to the role of CshA in *pdh* mRNA turnover. Interestingly we previously showed that RNase Y, when devoid of its membrane anchor, was able to suppress the *ΔcshA* growth defect [30]. There, we hypothesized that RNase Y would access more efficiently some mRNAs when not anchored to the membrane, therefore compensating for the CshA requirement for degradation. We therefore analyzed the *pdh* mRNA accumulation in a *ΔcshA/rny*$^{A2−24}$ double mutant grown on plates at 25°C by RT-qPCR and found that, whereas the *pdh* level stays unchanged in *pdh* or *ahrC* mutants, the *rny*$^{A2−24}$ mutant restores the *pdh* mRNA close to *wt* level (Fig 6F). Altogether these results suggest that the requirement of CshA for efficient *pdh* mRNA decay could explain at least in part its cold sensitivity.

## Acetate supplementation inhibits growth at 25°C in MH media

Our data suggest that over-expression of *pdh* interferes with accurate BCFA synthesis. In respect to this hypothesis, a recent study proposed that the membrane composition is influenced by carbon flow and in particular acetyl-CoA flow [31]. It showed that addition of acetate in the medium leads to a decrease in BCFA, probably due to an overflow of acetyl-CoA towards SCFA synthesis, which we believe also occurs in the *ΔcshA* strain. The authors also showed that addition of acetate also leads to increased staphyloxanthin production, known to have a rigidification effect on the membrane. We thus tested the effect of acetate supplementation both in *wt* and *ΔcshA*, but also in their respective derivates devoid of the capacity to produce staphyloxanthin, by the deletion of the *crt* operon, to exclude any effect of its product. Fig 7A shows that, whereas addition of acetate has a minor effect on growth at 37°C, the growth at 25°C of both the *wt* and *ΔcshA* strain is markedly affected, resulting in a pronounced growth delay. Results are similar in the *Δcrt* derivates excluding any major effect of staphyloxathin production in the growth delay (Fig 7B), which is in accordance with a recent study where the

**A.**

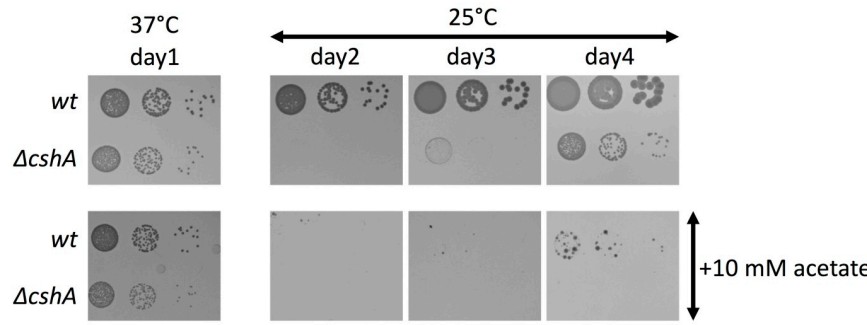

**B.**

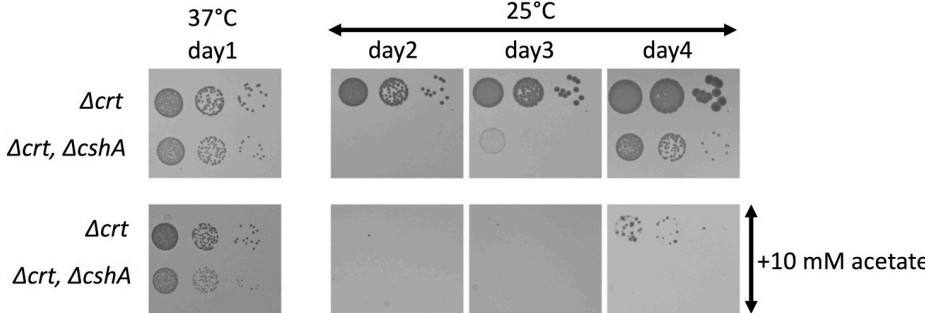

**Fig 7. Acetate supplementation alters cold growth of the *wt* strain. (A)** Over-night cultures of *wt* (PR01) and *ΔcshA* (PR01-ΔcshA) were serially diluted and spotted on MH plates containing uracil and, when indicated, 10mM Na-acetate and incubated at 37˚C or 25˚C as indicated. **(B)** Staphyloxanthin does not influence cold growth. Over-night cultures of *Δcrt* (SVK109) and *Δcrt/ΔcshA* (SVK110) were serially diluted and spotted on MH plates containing uracil and, when indicated, 10mM Na-acetate and incubated at 37˚C or 25˚C as indicated.

respective impact of staphyloxanthin and BCFA production on growth was evaluated and showed that the second had a major impact whereas the first affected growth to a minor extent [31,32].

## Effect of acetate on different *Staphylococcus aureus* strains

We tested whether acetate sensitivity at 25˚C observed above was a specific characteristic of the strain used in this study or common to different *Staphylococcus aureus* lineages. Fig 8 shows that acetate supplementation impedes growth of almost all strains tested, with a minor, if any, effect on strain 8325.4. Interestingly the acetate supplementation growth defect at low temperature was suppressed by addition of triclosan as observed for the *ΔcshA* strain, suggesting a similar mechanism. Astonishingly, we noticed a large variability of cold growth between strains in MH even without acetate addition, with a marked slow growth for USA300 JE2 and Newman strains. More strikingly this slow growth phenotype was also suppressed when triclosan was added in the plate raising the question of a defect of BCFA content in these strains. It is to note that cold growth defect by addition of acetate is dependent on the media. Indeed, acetate addition in LB or TSB does not lead to a pronounced growth defect as observed in MH, with smaller colonies already appearing after 2 days of incubation for most of the strains (S2

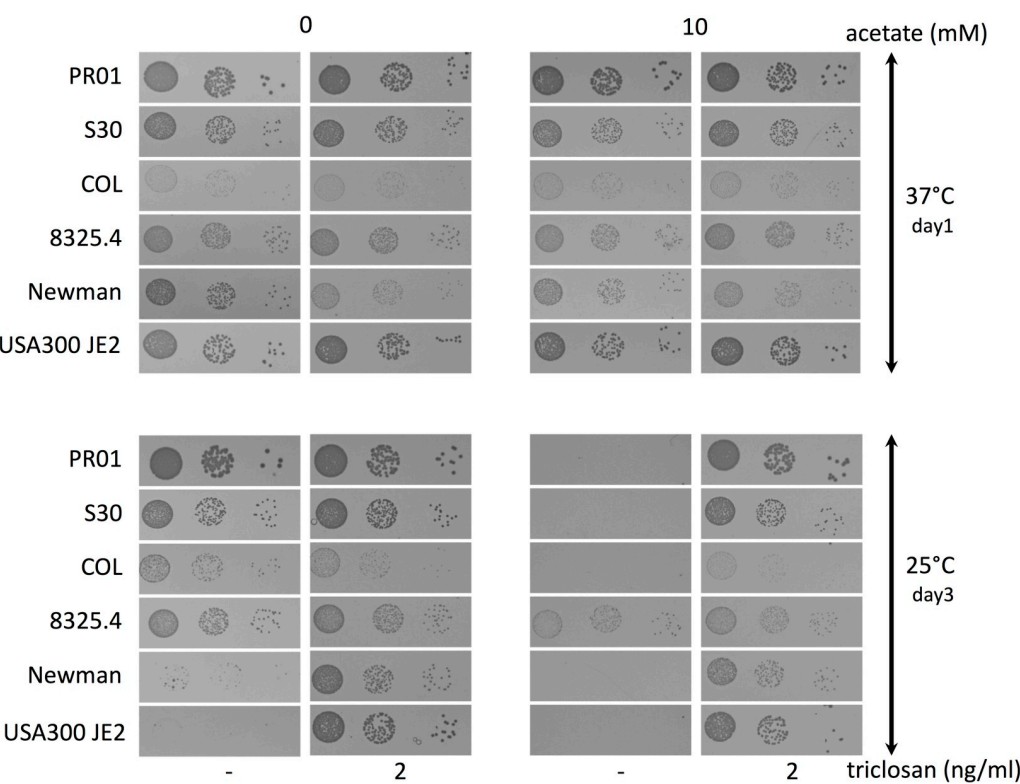

**Fig 8. Acetate supplementation alters cold growth of various *S. aureus* strains.** Over-night cultures of various *S. aureus* strains, PR01, S30, COL, 8325.4, Newman and USA300 JE2 were serially diluted and spotted on MH plates containing uracil and, when indicated, 10 mM Na-acetate and/or 2 ng/ml of triclosan and incubated at 37˚C or 25˚C.

Fig). The Newman and USA300 JE2, in contrast to what observed in MH, grew correctly at low temperature in both LB and TSB. However, in contrast to the other strains, they are sensitive to acetate supplementation, in particular USA300 JE2. Moreover, this defect was compensated by triclosan addition which underlines a correlation between the efficiency of growth at low temperature in MH and sensitivity to acetate in other media.

## Suppressor mutations from the *ΔcshA* screen suppress the acetate mediated growth defect at 25˚C

As shown above, addition of triclosan which suppresses the *ΔcshA* cold growth defect, also restore the slow growth due to acetate supplementation of the *wt* strain. We therefore tested whether mutations that suppress *ΔcshA* cold growth could also compensate for growth on acetate at 25˚C. For this purpose, various mutations identified in the screen were constructed in the *wt* strain. We first tested the above described mutations: 4 ACC mutants (*accC*$^{M385V}$, *accD*$^{A164V}$, *accD*$^{A164E}$, *accD*$^{F253V}$), the *fabD*$^{Q164L}$, *pdh*$^{A2P}$, *ahrc*$^{R44C}$ and the *rny*$^{Δ2–24}$, which all indeed counteracted the defect of growth observed with acetate (Fig 9A). We also tested whether the over-expression of BCKD enzymes, the AhrC$^{R44C}$ mutant or the FapR protein, all shown above to compensate *ΔcshA* cold growth, also improve the growth of the *wt* strain in acetate supplemented media at 25˚C, which indeed was the case (Fig 9B). In addition to these mutations, the screen identified other genes, not studied further here, but for which we wanted to see whether their mutations were also able to compensate cold growth on acetate which would indicate a related mechanism of suppression. Our results showed that all mutations

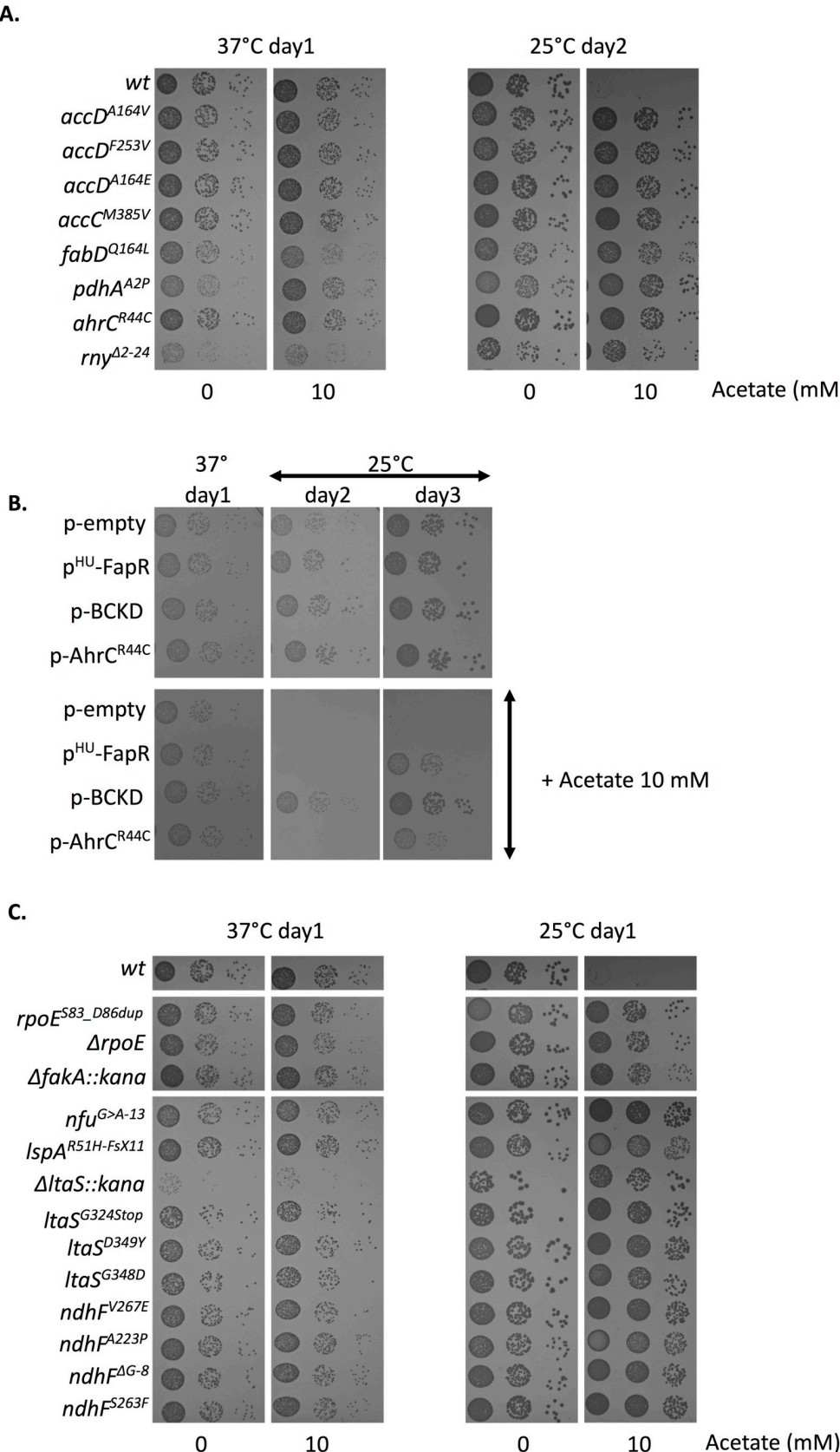

**Fig 9. Cold growth defect by acetate supplementation is compensated by *ΔcshA* suppressor mutations. (A)** Over-night cultures of *wt* (PR01), *accD*$^{A164V}$ (SVK131), *accD*$^{F253V}$ (SVK132), *accD*$^{A164E}$ (SVK133), *accC*$^{M385V}$ (SVK134), *fabD*$^{Q164L}$ (SVK86), *pdh*$^{A2P}$ (SVK88), *ahrC*$^{R44C}$ (SVK128) and *rny*$^{Δ2-24}$ (PR01-03) were serially diluted and spotted on MH plates containing, when indicated 10mM acetate and incubated at 37°C and 25°C. **(B)** Over-night cultures of transformants of *wt* (PR01) strain with empty vector (pCN47) or plasmid expressing FapR (pVK102), BCKD (pVK197) or AhrC$^{R44C}$ (pVK176), were serially diluted and spotted on MH plates containing erythromycin, and 10mM acetate when indicated, and incubated at the indicated temperatures. **(C)** Over-night cultures of *wt* (PR01), *rpoE*$^{S83\_D86dup}$ (SVK41), *ΔrpoE* (SVK39), *ΔfakA::kana* (SVK47), *nfu*$^{G>A-13}$ (SVK115), *lspA*$^{R51H-FsX11}$ (SVK116), *ΔltaS::kana* (SVK43), *ltaS*$^{G324Stop}$ (SVK119), *ltaS*$^{D349Y}$ (SVK120), *ltaS*$^{G348D}$ (SVK123), *ndhF*$^{V267E}$ (SVK124), *ndhF*$^{A223P}$ (SVK125), *ndhF*$^{ΔG-8}$ (SVK126), *ndhF*$^{S263F}$ (SVK127), were serially diluted and spotted on MH plates containing, when indicated, 10mM acetate and incubated at 37°C and 25°C.

tested with no exception compensated indeed the growth of the otherwise *wt* strains when grown at 25°C on acetate (Fig 9C). The mutations are in the *rpoE*, *fakA*, *lspA*, *ltaS*, *nfu*, and *ndhF* genes. The *rpoE* gene encodes the additional δ-subunit of the firmicute RNA polymerase [33]. We constructed the *rpoE*$^{S83\_D86\ dup}$, as selected in the screen as well as a full deletion of the *rpoE* gene. Both were able to compensate the cold growth on acetate (Fig 9C), however we have not identified the transcriptomic changes that could account for the observed pheno-types. The *fakA* gene, that encodes a fatty acid kinase, was found inactivated multiple times in the screen (S1 Table). FakA, along with its lipid binding partners, phosphorylates free fatty acids and is essential for both incorporation of exogenous fatty acids and fatty acid turnover [18,34]. It was recently reported that its inactivation alters acetate metabolism by an unknown mechanism [35], which would explain its occurrence in the screen as favoring the redirection of acetyl-CoA flow towards acetate production rather than SCFA. The *ltaS* gene encodes the lipoteichoic acid synthetase responsible for lipoteichoic acid (LTA) synthesis [36]. This gene was found mutated in 12 strains of the genetic screen, and most mutants inactivated the syn-thesis of LTA as shown by the absence of LTA analyzed by western blot (S3 Fig). We con-structed both a complete deletion of the *ltas* gene or introduced stop codon or point mutations in the binding pocket of LtaS (*ltaS*$^{G324stoP}$, *ltaS*$^{D349Y}$, *ltaS*$^{G348D}$) as found in the screen and showed that all restored the 25°C growth on acetate (Fig 9C). The *lspA* gene encodes the lipo-protein-specific type II signal peptidase [37]. The mutation introduced a frameshift at the beginning of the coding sequence indicating that *lspA* is inactivated, and that lipoproteins are not entirely matured with the signal peptide still remaining. Inactivation of *lspA* or *ltaS* likely have major and various impacts on membrane properties as lipoteichoic acid and lipoproteins are abundant components of the membrane, but the particular reason for their presence in the screen has not been studied further. The *nfu* gene encodes a Fe-S cluster carrier [38,39]. The mutation is upstream of the coding region in the Shine Dalgarno sequence (AGGAGA to AAGAGA) likely decreasing the expression of the Nfu protein. All proteins requiring Fe-S cluster are possible interesting targets to explain the observed phenotype. It is interesting to note that Nfu and SufT, another Fe-S cluster carrier, were suggested to be important for LipA activity, which requires a Fe-S cluster and is, in turn, essential for lipoic acid synthesis, an essential cofactor of PDH [40]. It is tempting, although purely speculative, to envisage that this mutation decreased PDH activity. The last gene tested is *ndhF*, encoding a NADH dehydroge-nase responsible for the recycling of NAD$^+$ from NADH. Of the 5 mutations in this gene one is the insertion of a mobile element between the promoter and the coding sequence, whereas one is a change in the Shine Dalgarno sequence, suggesting that decreased activity of NdhF is responsible for the observed phenotype. Such a decrease, leading to an increased NADH/NAD$^+$ ratio, would therefore have many consequences on metabolism but one interesting hypothesis is that it regulates the activity of PDH which is inactivated by NADH [41]. It is also to note that NdhF contains a Fe-S cluster and maybe the important target for the *nfu* mutation.

Whatever the specific mechanism for all selected mutants, our data suggested that these genes all participate to the same mechanism, i.e., modifying membrane properties. In agreement with this, analysis of fatty acid profiles in two of the *ΔcshA* suppressor strains containing *rpoE* or *fakA* mutants, respectively, showed an increased BCFA production compared to the parental *ΔcshA* (Fig 4B). The above results indicate that acetate supplementation on *wt* cells mimics the effect of a *ΔcshA* deletion, strongly supporting our hypothesis that cold sensitivity of the helicase mutant is at least partially due to SCFA/BCFA ratio imbalance, caused by *pdh* mRNA accumulation, which in absence of CshA is no longer degraded efficiently.

## Discussion

### Cold sensitivity of *ΔcshA* strain is in part due to imbalance of SCFA versus BCFA

Our previous work showed that the inactivation of the gene encoding the DEAD-box RNA helicase CshA from *S. aureus* results in a growth defect at low temperatures [8,9]. This is reminiscent of mutations in two of the five *E. coli* RNA helicases, CsdA and SrmB, which render the bacteria cold sensitive, if absent. Both of these proteins were shown to be involved in ribosome biogenesis and a suppressor screen in a *ΔsrmB* context revealed mutations in the 5S and 23S rDNA [13]. Moreover, it has been suggested that CshA in *B. subtilis* may be involved in ribosome biogenesis, since its absence led to a decrease in mature ribosome and a relative increase of the 30S subunit [3]. Since we and others have shown that CshA associates with RNases in *S. aureus* and *B. subtilis* and is involved in RNA turnover [4,6,9], the aim of this study was to decipher the reason of the cold sensitivity caused by the absence of CshA. We therefore have undertaken a large genetic screen and selected 82 spontaneous and independently occurring suppressor mutations. The screen was incredibly powerful and enabled us to pinpoint one major physiological pathway affected: two-thirds of the mutations are present in genes involved in membrane synthesis. This immediately suggested that the helicase mutant might be, at least in part, deficient in adapting the membrane composition, when shifted to lower temperatures. The mutations in the *ahrC* gene directly pinpointed towards a requirement of increased BCFA production to compensate the *ΔcshA* growth defect. We found that the two *ahrC* mutants led to increased expression of the *bckd* genes located downstream of the *ahrC* gene which controls its own expression, whereas in contrast, overproduction of wt *ahrC* drastically decreased the ability of the *wt* cells to grow in the cold. We confirmed that BCKD over-expression by itself compensated the growth defect of the *ΔcshA* strain by expressing the four genes from a multicopy plasmid. Proper function of the membrane at low temperature requires adaptation involving mostly fatty acid composition changes, allowing incorporation of lower melting point molecules. In *E. coli*, this is achieved through increased production of unsaturated fatty acid. In some Gram-positive bacteria, such as *Bacillus subtilis* but not in *Staphylococcus aureus*, desaturase exists and is proposed to be involved in short-term adaptation [42,43] whereas the major response for long-term adaptation relies on increased production of branched-chain fatty acids and decreased chain-length [15]. We therefore performed a fatty acid composition analysis in the *wt* and *ΔcshA* strain at different temperatures. As expected the SCFA content in the *wt* strain decreased at lower temperature. In sharp contrast to the *wt* situation, the SCFA content stayed unchanged between 37˚C and 25˚C in the *ΔcshA* strain. Altogether these experiments indicated that the cold sensitivity of the *ΔcshA* strain is due at least in part to its inability to accommodate the BCFA content at low temperatures. A conclusion supported by the fact that all suppressor mutants tested restored BCFA content close to *wt* strain level.

## *pdh* mRNA accumulation in *ΔcshA* could explain part of the cold sensitivity

To find the molecular basis explaining the defect of BCFA observed in a *ΔcshA* strain, we took advantage of our previous study where overall mRNA decay was measured in the helicase mutant compared to the *wt* [9]. From this analysis two operons, *fapR* and *pdh*, retained our attention because they were decaying slower in *ΔcshA* and both were found mutated in our genetic screen. The first operon encodes, in addition to FapR, the three proteins PlsX, FabD and FabG, involved in fatty acid or phospholipid synthesis. Upregulation of the FapR regulon in *B. subtilis* leads to changes in fatty acid profile and cold sensitivity [23]. However, the hypothesis that the *ΔcshA* defect could be due to increased expression of *fapR* operon was invalidated as the deletion of *fapR* does not counteract the suppressor effect of *acc* or *pdh* mutations, nor did it led to cold sensitivity of the *wt* strain in contrast to the observation made in *B. subtilis* [23]. Our second candidate, the *pdh* operon, in which we found 5 different mutations, appears as an attractive candidate in light of a recent study, which showed that inactivation of the pyruvate dehydrogenase leads to a marked increase of BCFA [29]. We therefore hypothesized that its over-expression could result in a decreased BCFA content. We showed that, as expected, the *pdh* operon was over-expressed in the *ΔcshA* mutant and that its over-expression from a multicopy plasmid in the *wt* strain strongly affected the growth, in particular at low temperature. Importantly this effect was alleviated by the *ΔcshA* suppressor mutation in the *fabD* gene or by addition of triclosan, which favors the growth of *ΔcshA* at low temperature. Acetyl-CoA is used at two steps in fatty acid synthesis: first by the ACC to produce malonyl-CoA and second by FabH, using it as the primer of the FASII cycle to produce SCFA. The fact that we observed no change in the transcriptional activity of the *fapR* repressor in the 5 *pdh* suppressor strains, argues for a major effect of the mutation on the step carried out by FabH. Interestingly, the 2 *pdh* suppressor mutants tested for fatty acid analysis showed the most drastic changes, reducing the level of SFCA below the *wt* level. The effect was even more dramatic when one of the mutations was transferred to the *wt* strain, going from about 10% of SCFA at 37˚C to 6% at 25˚C (Fig 6B). These experiments showed that reducing PDH activity could have tremendous effects on BCFA versus SCFA content and that *pdh* over-expression is highly deleterious for growth.

## Acetate supplementation leads to the same defect as deletion of *cshA*

Our data indicate that acetyl-CoA flow towards SCFA production is exacerbated in the *ΔcshA* strain, leading to an improper ratio of SCFA versus BCFA. This conclusion is in agreement with a recent study that established a link between central metabolism and membrane composition and showed that supplementation of the media with acetate leads to increased production of SCFA [31]. A key experiment of our study was the observation that acetate supplementation has a deleterious effect on growth at low temperatures. More importantly, this effect was suppressed in various conditions that we showed here to suppress the *ΔcshA* cold growth defect, which strongly reinforces our conclusion. Regulation of fatty acid composition is far from being well understood in *S. aureus*. Our study contributes to a growing body of evidence linking fatty acid composition to central metabolism, with the PDH complex and acetyl-CoA flow at the center of this regulation. It enlightens its importance since its perturbation could actually have major effects on BCFA content which in turn leads to a marked effect on cold growth. We also observed large differences between media or strains, which probably underlines the complexity of the interplay between central metabolism and fatty acid content. To be fully understood, this would require the combination of metabolomic, transcriptomic and fatty acid profile analyses in various strains and growth conditions, which is well beyond

the scope of this study. It nevertheless pinpoints the fact that these aspects need be taken in consideration when studying these processes.

## Mutations in genes involved in fatty acid metabolism restored *ΔcshA* growth at low temperature

In addition to mutations in *pdh*, we found that mutations in FASII genes or the addition of substance like triclosan favored the growth of *ΔcshA* at low temperatures. Our data indicate that whereas over-expression of FASII genes has no effect on cold growth in contrast to what is observed in *B. subtilis*, their downregulation, either transcriptionally or by mutation, restored *ΔcshA* growth, indicating that the activity of the FASII enzymes by itself is important for setting the BCFA content. How this is achieved has not been directly investigated in this study, however, we have performed an extended fatty acid analysis with the *fabD* mutants, both in the *ΔcshA* and in an otherwise *wt* background (S4 Fig). Lower production of SCFA in *fabD* mutants is already observed at 37°C (14%), suggesting that decreased production of malonyl-ACP favors BCFA precursor-priming by FabH by an unknown process. Interestingly, decreasing the temperature does not lead to further decrease in SCFA, and level stays similar to the 37°C condition. This is different of what is observed in the *pdh* mutants which have lower level of SCFA at 37°C (around 10%) and decrease them further at 25°C down to about 6%. It thus appears that the *fabD* mutant does no longer respond to the temperature whereas the *pdh* mutant still does, even if the BCFA content is already high. In addition to FASII genes, another gene involved in fatty acid metabolism, *fakA*, was found mutated in the screen. The high occurrence of mutations (in 35 strains) in *fakA* probably reflects two biases of the genetic screen: the efficiency of suppression in genes whose mutations have no obvious fitness cost and multiple ways to inactivate a gene. It is important to note that this screen therefore provides a large collection of mutants that could be interesting in a study aiming at elucidating the structure-function of the FAK complex. The FAK complex, which is constituted of the kinase FakA and the lipid binding partners FakB1 or FakB2 that present the fatty acid to the kinase, plays an important function in acquisition of fatty acid from the host, as phosphorylation of free fatty acid make them competent for incorporation in phospholipid synthesis pathway [18]. It was also shown to promote phosphorylation of endogenous free fatty acid and thereby promoting fatty acid turnover in *S. aureus* [34]. A recent report and particularly interesting study in light of our own showed that inactivation of *fakA* impacts the acetate switch, resulting in high secretion of acetate during exponential growth [35]. We can thus propose that *fakA* mutations help *ΔcshA* growth through the redirection of acetyl-CoA flow away from SCFA synthesis. To date it remains unknown how the absence of FakA could have such an effect on metabolism, which constitutes a highly relevant topic in the field in the near future, but it exemplified another link between fatty acid and metabolism.

## Cold sensitivity and function of CshA in mRNA decay

In the present study, we proposed that part of the cold growth defect of the *ΔcshA* mutant is due to its implication in *pdh* mRNA turnover. Along with the mRNA decay defect hypothesis, we previously described a suppressor allele of *rny* encoding RNase Y that can restore growth at low temperature [30] and we showed here that it restores *pdh* mRNA levels. So far we have shown that the two major phenotypes associated with the inactivation of *cshA*, (i.e., decrease in biofilm formation and cold sensitivity) could be related to its function in mRNA decay, showing the importance of this process in the adaptation to changes in growth conditions. However, our studies do not exclude an additional role of CshA in ribosome biogenesis and more experiments will be needed to identify the precise sequences or structures targeted by CshA.

## Materials and methods

### Growth of strains and media

All strains were grown in Mueller-Hinton broth (Becton-Dickinson), or when indicated in LB or TSB (Becton-Dickinson). Uracil to 20 mg/l, was added in all conditions as the parental *wt* strain used for this study (PR01) is *ΔpyrEF*. 10 mg/l erythromycin or 1 to 4 ng/ml triclosan (Sigma-Aldrich) was added when indicated. 10mM Na-acetate (EMD-millipore corp) was first added in conjunction to 20 mM MOPS pH7.4 to avoid acidification of the media but this was shown to be unnecessary.

### Strains and plasmids

Suppressor strains were selected by platting 50 μl of ON cultures of *ΔcshA* strain grown at 37˚C in MH medium, our standard lab medium, on MH plate at 25˚C, a temperature where the *ΔcshA* mutant still grows but with a marked delay. Candidates were further purified once at 25˚C, to selected true positive candidates and once at 37˚C, to evaluate their capacity to grow at higher temperature. The various mutants on the genome were reconstructed as described in [44]. Plasmids and strains are described in S1 Table.

### Variant calling

Genomic DNA were extracted using the DNeasy Blood and Tissue Kit (Qiagen), preceded with a lysis step using lysostaphin to 200ug/ml in TE, during 15 min at 37˚C.

To identify the mutations in the genomes of the 82 *ΔcshA* cold-suppressing strains, we carried out whole genome sequencing with Illumina HiSeq 2500 technology (Fasteris SA). In order to optimize library preparation costs, the DNA of 6–12 samples were pooled together at equal concentration and prepared as one single library. Using this pooling strategy, a mutation in one strain is expected to affect 8%-16% of the reads at the concerned genomic locus, which is far above the expected sequencing error of the instrument (~1%). Regarding sequencing depth, we targeted ~3 millions paired-end reads of size 2x125bp per strain, which approximately corresponds to a coverage of 300x.

For the identification of the mutations, we built an analysis pipeline on a *de novo* assembly strategy that can detect big rearrangements of the genome. More specifically, we used the *de novo* assembly software SGA [45] to filter the reads of low quality and trim the adapters, to perform kmer-counting in the reads, and to generate contigs by *de novo* assembly without simplification of the assembly graph. In all the processes we used k-mers of 61 nucleotides (61-mers). More specifically to put in evidence mutations (single nucleotide polymorphisms, insertions, deletions or big rearrangements) in a sequenced pool of DNA, we used 61-mers that are specific of the sequenced reads when compared to the reference genome, *S. aureus* strain SA564 (RefSeq assembly GCF_001281145.1). To further reduce false positive matches produced by sequencing errors, we additionally required the 61-mer to be seen at least 10 times in the sequenced reads, and 10 times in the reverse-complemented reads. Then the resulting pool-specific 61-mers were used to identify the contigs of the *de novo* assembly that carry the mutations. These mutated contigs were mapped on the reference genome with the software BWA [46]and converted to BAM format for manual investigation in the genome browser IGV.

A custom R script additionally processed the mapping results and generated a report summarizing several statistics that helped in manual curation of the output. In particular, the report contained for each mutated contig the length of the contig, the number of 61-mers that have allowed to call this mutated contig, the average number of occurrences of the 61-mers in the sequenced reads, whether the contig aligned on the reference genome, in case of alignment

the positions (potentially multiple) where it aligned and genes located nearby, and an analysis of the CIGAR string to identify the type of mutation (e.g. SNP, insertion, deletion, rearrangement). The calling pipeline has been automatized and the code is freely available at (https://gitlab.unige.ch/LinderLab/evar) and the software has been packaged as a docker container (plinderlab/evar).

Data availability: the DNA sequencing data of the *ΔcshA* suppressors have been deposited on GEO database with accession number GSE133013.

## RNA extraction

RNA was extracted using the RNAsnap method [47] from cultures harvested in mid-exponential phase at an OD600 of ~0.4 or from cells scrapped from solid agar media. For decay analysis, rifampicin to 400 ng/ml was added at OD600 of ~0.4 and aliquots were withdrawn at the indicated times. Cultures were immediately mixed with large volume of ice-cold ethanol/acetone (50% v/v), thereby immediately blocking further RNA decay as well as killing the cells. Cells were broken using 0.1mm zirconia/silica beads (Biospec Product) in MP Biomedicals Instrument FastPrep-24, 3X30sec at maximal speed. RNA was either directly used for Northern blot or further purified for RT-qPCR by treating 5 to 10 μg of total RNA with RQ1 DNase (Promega), reducing final concentration of formamide to 5% in the reaction, and incubated 30 min at 37˚C. RNA was then cleaned-up using RNA Clean & Concentrator-5 kit (Zymoresearch) with an additional DNase step on column as specified by the manufacturer.

## RT-qPCR

mRNA levels were quantified by RT-qPCR using 2,5 ng of total RNA per reaction and the Promega kit (GoTaq-1step RT-qPCR system) and the following cycling condition: 42˚C 15min/95˚C-10 min/(95˚C-10sec/60˚C-30sec/72˚C-30sec)X40-melting curve 60–95˚C in a CFX96 C1000Thermal Cycler (BioRad). Oligonucleotides are indicated in S1 Table. 16S was used as the reference gene [48].

## Northern blot

Equal amounts of total RNA in each lane were separated on 1.2% agarose/formaldehyde gel and MOPS buffer, transferred on HybondN$^+$ membrane (Amersham) by capillarity blotting and crosslinked using UV stratalinker 2400 (Stratagene). Loading was controlled by methylene blue coloration of the membrane, enabling to position 16S and 23S rRNA migration used as size markers. DNA oligonucleotides radiolabeled using T4PNK kinase and (γ32P)-ATP, were hybridized to the membrane in ExpressHyb hybridization solution (Clontech, Mountain View, CA, USA) over-night at 37˚C and washed. Quantification was done by detection of the signal using a Typhoon FLA7000 phosphoimager (General Electric). Oligonucleotides are indicated in S1 Table.

## Fatty acid profile

Over-night culture was diluted and plate to obtain between 4000 to 8000 bacteria on MH containing uracil large plate (diameter 150 mm) and incubated over-night at 37˚C or 2, 3 or 5 days at 25˚C for *wt*, suppressor strains or *ΔcshA* strain respectively. Cells were collected in PBS, wash twice, and lyophilized. Fatty acid analyses were carried out by the Identification Service of the DSMZ, Braunschweig, Germany, using an Agilent 6890N gas chromatograph and version 6.1 of the MIDI Inc Sherlock MIS software. The method of preparation of the samples is the standard method given in MIDI Technical Note 101.

## Western Blot

LTA was extracted from over-night cultures normalized based on $OD_{600}$ readings. Samples were centrifuged and resuspend in 300 μl of PBS, lysed using 0.1mm zirconia/silica beads (Biospec Product) in MP Biomedicals Instrument FastPrep-24, 3X30sec at maximal speed. Pellets were resuspended in 100 μl of 2X SDS-page loading buffer, incubated at 95°C for 30 min, centrifuged at 17,000 x *g* for 5 min. 10 μl aliquots were separated on 15% SDS-PAGE gels and subsequently transferred to a PVDF membrane. LTA was detected using the monoclonal poly-glycerolphosphate-specific LTA antibody (1:250; Clone 55 from Hycult Biotechnology) and the anti-mouse IgG-Peroxidase antibody (1:5000, Sigma) and blots were developed by enhanced chemiluminescence (ECL-prime Amersham-GE-healthcare).

## Supporting information

**S1 Fig. *fabD* mutation but not *pnkB* mutations suppresses *ΔcshA* cold growth.** Over-night cultures *wt* (PR01), *ΔcshA* (PR01-ΔcshA), *ΔcshA/pnkBS523F* (SVK87), *ΔcshA/fabDQ164L* (SVK92) and *ΔcshA/fabDQ164L/ pnkBS523F* (C53) strains were serially diluted and spotted on MH plates and incubated at 37 C or 25 C.
(TIF)

**S2 Fig. Effect of acetate in LB and TSB media.** Over-night cultures of various *Staphylococcus aureus* strains, PR01, S30, COL, 8325.4, Newman and USA300 JE2 were serially diluted and spotted on LB (top panels) or TSB (bottom panels) plates containing when indicated 10 mM Na-acetate and 2 ng/ml of triclosan and incubated at 37°C or 25°C as indicated.
(TIF)

**S3 Fig. *ltas* mutants impairs LTA synthesis.** LTA detection by western blot using anti-LTA antibodies in *wt* (PR01), *ΔcshA* (PR01-ΔcshA), and the 12 suppressor strains containing mutations in *ltaS* (sup29, C34, sup22, C17, sup7, C22, C40, C52, C19, C62, C18 and C16). Note that no LTA detection was observed for all *ltas* mutants except in the C22 strain where LTA production seems to be lower down.
(TIF)

**S4 Fig. Fatty acid profile of *fabD* and *fakA* mutants.** Relative percentage of BCFA and SCFA from *wt* (PR01), *ΔcshA* (PR01-ΔcshA), *fabD*^Q164L^(SVK86), *ΔcshA/fabD*^Q164L^(SVK92 and C53), *ΔfakA::kana* (SVK47) and *ΔcshA/ ΔfakA::kana* (SVK48) 37 and 25°C, scrapped on plate. Mean and standards deviation are represented (n = 3 for *wt* and *ΔcshA* and 2 for the others, for *ΔcshA/fabD*^Q164L^ data from SVK92 and C53 were combined). See complete data set in S2 File.
(TIF)

**S1 Table. Bacterial strains, plasmids and oligonucleotides.**
(DOCX)

**S1 File. RT-qPCR Numerical data.**
(XLSX)

**S2 File. Fatty acids profile row data.**
(XLSX)

## Acknowledgments

We are grateful to Alexandra Gruss, Joshua Armitano, Stéphane Hausmann for continuous discussions and for comments on the manuscript.

## Author Contributions

**Conceptualization:** Vanessa Khemici, Patrick Linder.

**Data curation:** Vanessa Khemici, Julien Prados.

**Formal analysis:** Vanessa Khemici, Patrick Linder.

**Funding acquisition:** Patrick Linder.

**Investigation:** Vanessa Khemici, Bianca Petrignani, Benjamin Di Nolfi, Elodie Bergé, Caroline Manzano, Caroline Giraud.

**Methodology:** Vanessa Khemici.

**Project administration:** Vanessa Khemici, Patrick Linder.

**Software:** Julien Prados, Benjamin Di Nolfi.

**Supervision:** Vanessa Khemici, Patrick Linder.

**Validation:** Vanessa Khemici, Patrick Linder.

**Visualization:** Vanessa Khemici.

**Writing – original draft:** Vanessa Khemici.

**Writing – review & editing:** Vanessa Khemici, Patrick Linder.

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
