## [Decision Letter · Decision Letter 0]

17 Mar 2020

Dear Dr Linder,

Thank you very much for submitting your Research Article entitled 'The DEAD-box RNA helicase CshA is required for fatty acid homeostasis in Staphylococcus aureus' to PLOS Genetics. Your manuscript was fully evaluated at the editorial level and by independent peer reviewers.

First of all I would like to apologize for the long time it took for my decision, but the times were and are very crazy and I did not find time earlier.

As you will see from the reviewers comments, tall three highly appreciated your work and found that it is an original and remarkably well-documented study, which provides novel information on how membrane homeostasis is managed in Staphylococcus aureus.  However, they also identified some aspects of the manuscript that should be improved. Furthermore the Grammar should be improved throughout the text especially the use of plurals and prepositions. Also, a number of typos are present e.g. NADH deshydrogenase.

We therefore ask you to modify the manuscript according to the review recommendations before we can consider your manuscript for acceptance. Your revisions should address the specific points made by each reviewer.

[LINK]

Yours sincerely,

Carmen Buchrieser

Associate Editor

PLOS Genetics

Josep Casadesús

Section Editor: Prokaryotic Genetics

PLOS Genetics

Reviewer's Responses to Questions

**Comments to the Authors:**

Reviewer #1: Mutants in the DEAD-box RNA helicase CshA demonstrate a cold-sensitive phenotype. The paper represents a heavily genetic approach (appropriate for PLOS Genetics I would think) to understanding the underlying mechanism of this phenomenon. Supressor mutants were predominantly involved in the fatty acid biosynthesis pathway. The cshA mutant is defective in decreasing straight-chain fatty acids/increasing branched-chain fatty acids at low temperatures compared to the wild type. The suppressor mutants restore the wild type ratio of SCFAs to BCFAs at low temperatures. Inefficient degradation of pyruvate dehydrogenase mRNA in the mutant is proposed to play a major role in the phenomenon by favoring acetylCoA production that leads to unbalanced SCFA production. I find the work to be thorough and well written. It is not my policy to nit pick papers that I review. You do not need to convince this reviewer that staphylococcal membrane fatty acid homeostasis is an important and interesting topic.

Comments:

1. The switch in percentages between SCFA and BCFA is modest albeit reproducible apparently. Could the authors "drill down" a bit into their fatty acid analyses. For example could they look at individual fatty acid species and see how they are changing in the wild type, mutant, and suppressor mutants. They could calculate the average SCFA and BCFA carbon chain lengths to see if they shorten with lower temperature. Also is there branching switching to more anteiso (anteiso C15:0 most likely) at lower temperatures.

Reviewer #2: The manuscript “The DEAD-box RNA helicase CshA is required for fatty acid homeostasis in Staphylococcus aureus” by Khemici et al presents a comprehensive analysis of suppressors of a cshA mutant, leading to the discovery that CshA modulates expression of fatty acid synthesis (FASII) genes and dictates membrane properties. Cold-resistant suppressors of a cshA mutant mapped mainly to fatty-acid related mutations. Using different approaches, authors identify Pdh activity as the likely primary cause of cshA cold-sensitivity. Pdh synthesizes acetyl-CoA, a FASII precursor; when in excess, acetyl-CoA primes the synthesis of more rigid straight-chain fatty acids (it outcompetes branched chain acyl-CoA primers), explaining the cold-sensitive phenotype. The numerous cshA suppressor mutations, which affect steps in fatty acid metabolism, then make sense. Authors further show that acetate supplementation, which increases acetyl-CoA production, also leads to cold-sensitivity in a wild type background. Cold sensitivity due to acetate addition is alleviated in the cshA suppressor backgrounds. Several suppressor mutations were affected in membrane composition but were not clearly linked to FASII (e.g., fakA, ltaS, lspA); the present results suggest the existence of novel roles of the affected genes. Altogether, the manuscript shows that CshA activity contributes to regulating membrane rigidity by controlling Pdh-catalyzed synthesis of acetyl-CoA, the precursor of straight-chain fatty acids.

This study provides important findings in understanding S. aureus membrane physiology and its regulation. The numerous investigative approaches, rigor, and breadth of the study convincingly support authors’ main hypotheses and expand the present understanding of fatty acid regulation and membrane homeostasis. The “honest” presentation is a pleasure to read. Authors explain that the observed phenotypes are seen on solid, but not liquid medium, and are medium-dependent. This seeming limitation is actually a pointer for explaining metabolic regulation.

Please note the following suggestions/questions:

1- The main “drawback” of this in-depth study is in presenting so much data! Perhaps authors could make certain sections more concise, e.g., shortening the sections on acetate-induced cold sensitivity (starting l 386) would simplify reading. Interpretations of phenotypes might be more appropriate for Discussion.

2- The manuscript would benefit from including a model. Fig. 1 presents some results, but does not include subsequently presented information. It would be useful for readers to include the various pathways that compete for acetyl-CoA (the acetate connection as here, and also TCA, pigments and cell wall). The main concept (membrane fluidity as controlled by CshA) is exciting and could be depicted schematically. A shorter Discussion (there is redundancy with Results) with a summary Figure could better bring out the importance of the findings.

3- Does the ltaS mutation impact fatty acid composition as shown for other mutants?

4- L 122: Authors propose that “accumulation of the pdh mRNA in the DcshA strain at low temperature exacerbates the production of SCFA, at the expense of BCFA production”. Authors could discuss why their previous transcriptome study of a cshA mutant (Ref 9) did not detect changes in pdh.

5- L 268 Could the fact that solid, but not liquid medium, selects for suppressors indicate that the TCA cycle, which also uses acetyl-CoA, may be less active in colonies? Have authors compared cold-sensitivity of cshA in anaerobic vs aerobic conditions?

6- Authors state L 352 “lowering the FASII activity [with triclosan] compensates the over-expression of pdh”. How might this be explained?

7- L510 For proper membrane function:.. “in Gram-positive bacteria, increased production of branched-chain fatty acid and decreased chain-length are the major responses”. Note that some Gram-positive like B subtilis express a phospholipid desaturase, which may allow rapid membrane adjustments (DOI: 10.1128/JB.185.10.3228-3231).

8- Table S1: C40: ltaSF64L-FsX2, what is FsX2?; Line 235 should read ahrC., line 261 “higly”. Check for numerous typos.

Reviewer #3: The authors have an extensive history of influential manuscripts describing the roles performed by DEAD-box RNA helicases in the opportunistic pathogen Staphylococcus aureus. In the current manuscript the authors expand these studies by performing a genetic screen to identify suppressors of the cold growth defects exhibited by the delta-cshA mutant. Genome sequencing indicated that genes associated with fatty acid metabolism were altered in many of the isolated repressors. Extensive experimental analysis correlated the low temperature phenotype with CshA alteration of the membrane fatty acid composition at low temperature. These observations provide unique insights into bacterial perception and genetic/physiological response to low temperature stress associated with RNA helicase-mediated RNA decay and alteration of membrane composition.

Major comments

Mutation in the 82 suppressor strains suppresses the cshA cold sensitive phenotype allowing growth at low temperature. Many of these suppressors are in genes associated with fatty acid synthesis. Thus in these mutants a decrease in cellular ability to synthesize fatty acids, specifically BCFA, which be expected, synthesis of which would be required to make the membrane more fluid at low temperature. Thus should the cells not die at low temperature?

While reading the manuscript I frequently wondered if, overall, the gene/membrane composition observations could result from a secondary effect from a more global regulator/pathway whose effect is enhanced at 25oC? The authors also struggle with this possibility at a number of points in the text.

For example, 13 of the hits occur upstream of the formation of malonyl-CoA. Thus, overall could the results not be explained by a defect in malonyl CoA utilization since it is used in numerous other pathways.

Line 188; The authors should check if the assumption that malonyl-CoA levels are reduced in the mutant. A 2-fold reduction in transcript level may not generate a decrease in product levels that is significant enough to alter carbon flow. In the manuscript, results that change transcript levels by 2-fold are frequently observed.

Potentially associated with the above comments, I wonder why the link between suppressor mutation and fatty acid/phospholipid composition was only analyzed for a subset of the mutants in the figures. I understand the reasoning to focus on the mutants in the initial parts of the pathway however a substantial proportion of the hits were in fakA (35) and ltas (12).

With respect to fakA and ltas, how does their high prevalence in the data set fit into the overall conclusion that cshA indirectly affects membrane composition via pdh half-life since they act much below pdh.

Similar to the malonyl-CoA comment, could at least a portion of the results originate from fakA alteration of free fatty acid turnover? Although exogenous FAs do not shut off endogenous synthesis in Staphylococcus, could the results from the addition of exogenous fatty acids be insightful? In summary, further investigation of fakA and ltas could be insightful and further support the hypothesis.

Significant emphasis was placed on alteration of the effects caused by exogenous acetate. Were common effects associated with acetate addition the best way to prove the hypothesis? In addition, while the effect of acetate on growth of various Staph strains was extensively studied at 37 and 25oC including delta-cshA and delta-cshA-delta-crt and in the presence or absence of triclosan (Figs. 8 and S2), why was the suppression of cold sensitivity by the other suppressors identified in the screen not investigated?

Indicate why MH media was used in these studies. Importantly, was suppression of cold sensitivity also observed in: 1) liquid, 2) minimal media, and 3) rich media as well?

Similarly, why was 25oC chosen as the cold stress temperature? Are reduced or enhanced effects observed at higher and lower temperatures?

The answer to the media question above is provided on Lines 266-269, the suppressors did not suppress cold sensitivity in liquid media (which one?). While an interesting observation, it is also potentially disturbing re the overall hypothesis presented. It would be beneficial if the authors could investigate the causes of this observation further.

Additional comments

Lines 125-127: It is not clear how this work connects “central metabolism” with fatty acid homeostasis.

Figure 1. Include branch sites leading to the formation of BCFA and SCFA forms. Associated with the ACC step, add AccC (2) and AccD (3) to be consistent. Indicate where Lipoteichoic acid synthase (12) contributes to the depicted biosynthetic pathway(?).

Figure 2. Need an internal control for relative quantification in 2B.

Fig. 2D. Why was growth of cshB severely reduced by triclosan at both temperatures while wild type grew relatively normally?

Figure 3. Lines 243-249. Does ahrCN31stop also suppress the cold sensitivity through a dominant-negative effect in wild type?

How many of the identified suppressor genes convey suppression of the cold sensitivity to delta-cshA similar to ahrC and BCKD shown in this figure?

Figure 4 and S4. Are the SCFA values statistically significant, both within each strain and between strains? Have changes of a similar magnitude been observed in FA ratios in Staph (e.g. reference 26) or other Gram-positive bacteria during temperature shift?

Also, is an increase in BCFA of ~4% (wt 37 vs 25oC) or a decrease of ~7% (delta-cshA 37 vs 25oC) sufficient to generate a lipid membrane that is functionally more or less fluid, respectively?

Figure 5. Clarify why FapR regulation of target gene expression was observed to not be involved in suppression of low temperature sensitivity, as indicated by the authors. As I understand the system, fapR mutation should have suppressed the low temperature sensitivity, however it did not and thus apparently invalidates the hypothesis. I also do not see the utility or corresponding insight gained by introducing the accC/D and pdhA mutations into the delta-cshA-delta-fapR mutant. From the nature of the suppressor screen would it not be anticipated that the accC/D and pdhA mutations, in the absence of delta-frpR, would be expected to confer suppression of the cold sensitivity in a delta-cshA background, similar to the results shown for other suppressor genes in Figure 3? Are accC/D and pdhA even regulated by FrpR?

Am I missing something?

Why are fapR transcript levels elevated to a similar degree in delta-cshA and delta-cshA plus the 3 phdA mutants while other suppressors reduce fapR levels, as shown in Fig. 2? Is there a unique insight provided by the difference between the FapR effect on the two sets of mutants?

Lines 333-336; How is the pdh northern blot in Fig. 6D indicative of the enhanced accumulation of the entire pdh operon?

Lines 348-352; It is interesting that growth is inhibited by p-PDH at 37oC and the authors predict it reflects deregulation of another pathway that utilizes acetyl CoA.

Is this not evidence that the results could be explained by a defect in malonyl CoA utilization as it is used in numerous other pathways.

Fig. 6F. Would it be useful to repeat this experiment at 37oC? Possibly more insightful, how are fapR levels affected in the delta-rny strain?

Figure 7. A potentially useful addition to this or other figures would include results from the artificial alteration of membrane fluidity caused by exposure to staphyloxanthine or benzyl alcohol. Would pdh transcript levels change in response to these compounds?

Figure 8. Provide as Supporting Information?

Figure S2. How is LTA still formed in the C22 mutant?

“gram-negative” should be changed to “Gram-negative” in the entire document

Materials and Methods

Lines 640-644: Could the authors elaborate on the criteria used to: 1) select the initial suppressor colonies and 2) further purify candidates once at 25°C and why they were also purified at 37°C.

Line 707; Could a more suitable control gene other than 16S rRNA be used for RT-qPCR quantification? The quoted paper identifies 16S rRNA as suitable in stationary phase cultures.

Lines 709-721: Provide further details regarding the RNA loading control used for Northern analysis and how quantification of signal intensities was performed.

Lines 726-729: Provide further details re the loading controls, standardization, identification and relative abundance determination related to fatty acid composition identification.

Lines 730-740: Provide further details regarding how LRT was extracted and how loading was controlled for western blot analysis detection.

**Have all data underlying the figures and results presented in the manuscript been provided?**

Reviewer #1: Yes

Reviewer #2: Yes

Reviewer #3: Yes

PLOS authors have the option to publish the peer review history of their article (what does this mean?). If published, this will include your full peer review and any attached files.

Reviewer #1: No

Reviewer #2: No

Reviewer #3: No

---

## [Editor Report · Decision Letter 1]

15 Apr 2020

Dear Dr Linder,

We are pleased to inform you that your manuscript entitled "The DEAD-box RNA helicase CshA is required for fatty acid homeostasis in Staphylococcus aureus" has been editorially accepted for publication in PLOS Genetics. Congratulations!

Yours sincerely,

Carmen Buchrieser

Associate Editor

PLOS Genetics

Josep Casadesús

Section Editor: Prokaryotic Genetics

PLOS Genetics

Comments from the reviewers (if applicable):

**Data Deposition**

http://datadryad.org/submit?journalID=pgenetics&manu=PGENETICS-D-20-00207R1

**Press Queries**

---

## [Editor Report · Acceptance letter]

25 Jun 2020

PGENETICS-D-20-00207R1 

The DEAD-box RNA helicase CshA is required for fatty acid homeostasis in Staphylococcus aureus 

Dear Dr Linder, 

We are pleased to inform you that your manuscript entitled "The DEAD-box RNA helicase CshA is required for fatty acid homeostasis in Staphylococcus aureus" has been formally accepted for publication in PLOS Genetics! Your manuscript is now with our production department and you will be notified of the publication date in due course.

With kind regards,

Jason Norris

PLOS Genetics

On behalf of:
